# Natural Products Inhibitors of Monoamine Oxidases—Potential New Drug Leads for Neuroprotection, Neurological Disorders, and Neuroblastoma

**DOI:** 10.3390/molecules27134297

**Published:** 2022-07-04

**Authors:** Narayan D. Chaurasiya, Francisco Leon, Ilias Muhammad, Babu L. Tekwani

**Affiliations:** 1Scientific Platforms, Southern Research, Birmingham, AL 35205, USA; 2Department of Drug Discovery and Biomedical Sciences, College of Pharmacy, University of South Carolina, Columbia, SC 29208, USA; 3National Center for Natural Products Research, Research Institute of Pharmaceutical Sciences, School of Pharmacy, University of Mississippi, University, MS 38677, USA

**Keywords:** natural products, monoamine oxidases (MAO-A and -B), antidepressant, monoamine oxidase inhibitors, neurological disorders, neuroprotection, Parkinson’s disease, neuroblastoma

## Abstract

Monoamine oxidase inhibitors (MAOIs) are an important class of drugs prescribed for treatment of depression and other neurological disorders. Evidence has suggested that patients with atypical depression preferentially respond to natural product MAOIs. This review presents a comprehensive survey of the natural products, predominantly from plant sources, as potential new MAOI drug leads. The psychoactive properties of several traditionally used plants and herbal formulations were attributed to their MAOI constituents. MAO inhibitory constituents may also be responsible for neuroprotective effects of natural products. Different classes of MAOIs were identified from the natural product sources with non-selective as well as selective inhibition of MAO-A and -B. Selective reversible natural product MAOIs may be safer alternatives to the conventional MAOI drugs. Characterization of MAO inhibitory constituents of natural products traditionally used as psychoactive preparations or for treatment of neurological disorders may help in understanding the mechanism of action, optimization of these preparations for desired bioactive properties, and improvement of the therapeutic potential. Potential therapeutic application of natural product MAOIs for treatment of neuroblastoma is also discussed.

## 1. Introduction

Amine oxidases are a heterogenous group of enzymes that metabolize various monoamines, diamines, and polyamines produced endogenously for physiological functions or exogenous xenobiotic substances absorbed through dietary intake [1]. The amine oxidases are distinguished by their co-factor requirements and substrate specificities [2]. The flavin adenine dinucleotide (FAD)-dependent amine oxidases include mitochondrial monoamine oxidase A (MAO-A), monoamine oxidase B (MAO-B), and cytosolic polyamine oxidases (PAOs). Copper and topoquinone (TPQ)-dependent amine oxidases include plasma and tissue enzymes, also referred to as semicarbazide-sensitive amine oxidases (SSAOs) [3] (Figure 1). This review primarily focused on MAO-A and MAO-B due to their predominant role in oxidative deamination of endogenous neurotransmitter biogenic monoamines such as dopamine, epinephrine (EPI), and norepinephrine (NE) [4]. Changes in the physiological levels of these monoamines have been implicated in the pathophysiology of several neurological disorders.

Differential localization of MAO-A and -B in tissues determines their physiological functions. MAO-A and -B play an important role in deamination of biogenic amines in neural and peripheral tissues [1,5]. MAO-A is more predominant in peripheral tissues such as the intestine, liver, lungs, and placenta and protects the body by oxidation of biogenic monoamines amines in the blood or by preventing the entry of dietary monoamines into circulation [6,7]. MAO-B plays a similar protective role in the micro vessels, acting as a metabolic barrier. In the peripheral and central nervous systems, intra-neuronal MAO-A and -B protect neurons from exogenous amines, regulate the contents of intracellular amine stores, and control pharmacological actions of amine neurotransmitters.

**Figure 1 molecules-27-04297-f001:**
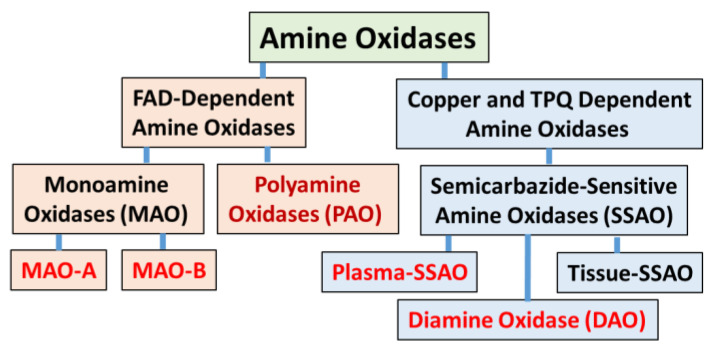
General classification of mammalian amine oxidases.

## 2. Structural Difference of MAO-A and MAO-B

MAOs, mitochondrial FAD-containing enzymes, exist as one of two functionally and structurally distinguishable isoforms, either MAO-A or MAO-B. Both isoforms are expressed in all tissues in different ratios. MAO-A is predominantly expressed in the heart, adipose tissue, and skin fibroblast and MAO-B is mainly found in platelets and lymphocytes, but the kidney, liver, and brain express both type of isozymes [8,9]. MAO-A and -B are distinguished by different substrate specificities and inhibitor sensitivities. MAO-A, selectively inhibited by clorgylin, preferentially deaminates norepinephrine and serotonin (5-HT). MAO-B, selectively inhibited by deprenyl and pargyline, metabolizes phenylethylamine and benzylamine. The three-dimensional crystal structures of human MAO-A and MAO-B with their specific inhibitors show significant similarities in their crystalline structure forms (Figure 2). However, important differences can be described in the oligomeric states and structures of substrate binding sites of MAO-A and -B [10,11]. In purified protein form, human MAO-B is dimeric, whereas human MAO-A exists in a monomeric form [12]. The active binding site of human MAO-B is a characterized by a hydrophobic dipartite cavity with a substrate entrance cavity (~290 Å^3^) connected with a larger substrate binding cavity (~420 Å^3^). The Ile-199 was identified as an important structural component of the enzyme active site because it plays an important role in the accessibility of the second substrate cavity. In the closed conformation, Ile-199 is physically responsible for the separation of two cavities, whereas bulky ligands are present to adapt an open conformation. The binding of clinical inhibitors (R)-deprenyl and rasagiline induces a midspan type of cavity pushing Ile-199 into the open conformation. The active site of human MAO-A differs from human MAO-B in the loop conformation region with residues 108–118 and 210–216. Both these regions are critical components of the enzyme active site [12]. The structural differences incorporating Ile-335 in MAO-A vs. Tyr-326 in MAO-B active sites are responsible for the differential susceptibility of MAO-A and MAO-B toward selective MAO inhibitors [11]. Both human and rat MAO-A contain 16 conserved residues surrounding the substrate and inhibitor cavity. Only 6 of the 16 residues differ between human MAO-A and MAO-B. Substrate and inhibitor selectivity of human MAO-A and MAO-B are determined by Ile-335 and Tyr-326, respectively [13,14]. The selectivity of reversible inhibitors is determined by the size and shape of the substrate and inhibitor cavity, which are related to Ile-335 and Phe-208 in MAO-A and Tyr-326 and Ile-199 in MAO-B. The differential inhibitor susceptibility of human MAO-A and -B is also attributed to their accommodation by the induced fit of Ile-335, which is similar to that observed for Ile-199 in MAO-B [15]. Neither human MAO-A nor MAO-B contain any disulfide bridges as determined by mass spectrometric analysis [8].

### 2.1. MAO Inhibitors (MAOIs)

MAO inhibition prevents the degradation of monoamine neurotransmitters in the target cells and reinstates their physiological levels required for normal physiological functions [16]. Derangements in physiological levels and homeostasis of biogenic monoamines lead to pathophysiological consequences. The most important effect of MAO inhibition is a rapid increase in the intracellular concentration of monoamines. A rise in the concentrations of monoamines leads to secondary adaptive consequences, including a reduction in biosynthesis of monoamines via an apparent feedback mechanism, which is most clearly demonstrated for the noradrenergic system [17]. Both reversible as well as irreversible MAOIs have been developed for treatment of disorders caused due to depletion of biogenic amines in the target cells [18,19,20]. The MAOIs are classified into three types (Table 1): (1) irreversible non-selective inhibitors, such as phenelzine and tranylcypromine, (2) irreversible selective MAOI drugs, such as selegiline and rasagiline, and (3) reversible selective MAO-A inhibitors (RIMAs), such as moclobemide [21,22].

### 2.2. Therapeutic Applications of MAO Inhibitors

The MAOIs are included in a group of drugs, commonly referred as thymoleptic drugs, that favorably modifies mood in serious mood disorders and is primarily prescribed for the treatment of clinical depression or mania [23]. The MAO-A inhibitors show efficacy for treating anxiety and depression while the inhibition of MAO-B appears to be effective for prevention and treatment of Parkinson’s disease [24,25]. MAOIs are also used as a medicine for controlling hypertension or treating depression and other neurological disorders. Psychiatric disorders, such as obsessive-compulsive disorder, somatoform pain, panic disorder, and schizophrenia, have been reported to occasionally respond to treatment with MAOIs.

The progressive death of dopaminergic neurons results in a deficiency of dopamine during the development of Parkinson’s disease (PD), a neurodegenerative disorder of the brain. PD is characterized by a combination of rigidity movements, lack of movements, tremors, and postural instability. Selegiline, a propargylamine, is an irreversible inhibitor of MAO-B which inhibits dopamine metabolism and has been used in the treatment of PD effectively [26,27]. However, the therapeutic utility of selegiline is compromised due to the generation of potential neurotoxic metabolites [28]. However, the neuroprotective effects of propargylamines in different neuronal models seem to be independent of inhibition of MAO-B. Glyceraldehyde-3-phosphate dehydrogenase (GAPDH), MAO-B, and/or other unknown proteins may be crucial targets in the survival of the injured neurons and may be crucial for the mechanism of neuroprotection by propargylamines. Further analysis of the mechanism(s) involved in the neuroprotective efficacy of MAO-B inhibitors may lead to the development of novel modalities for therapeutic applications. Selected MAO-A and -B inhibitors are listed in Figure 3 and Table 1. Historically, non-selective MAOIs have had in their structure a hydrazine group. The irreversibility of several MAOIs is caused by the triple bond scaffold which interacts directly with the active site and forms a covalent bond, irreversibly inactivating the enzyme (see Table 1) [2,5].

**Table 1 molecules-27-04297-t001:** Monoamine oxidase inhibitors (MAOIs) classified on the basis of their selectivity [28,29].

Inhibitor Type	MAO-A	MAO-B	Non-Selective
**Irreversible inhibitors**	Clorgyline; Lilly 51641	(-) Deprenyl Lilly 54781 MDL 72145 AGN 1133 AFN 1135 Rasagiline Pargyline Selegiline	Phenelzine Tranylcypromine Isocarboxazid Nialamide Iproniazid Safrazine Metfendrazine
**Reversible inhibitors**	Harmaline Amiflamine Cimoxatone Moclobemide Brofaromine Ro 11-1163 Toloxatone MD 780515 FLA 336(+)	Safinamide	

## 3. MAO Inhibitors in Clinical Development

Several new MAOIs are currently under development and clinical evaluation for the treatment of neurological diseases [26]. Impaired monoamine-mediated neurotransmission is directly connected to neurological disorders such as depression and anxiety as well as susceptibility to stress [27]. The majority of the antidepressants presently used are designed to control one or both of the most important neurotransmitters of the brain (DA and serotonin (5HT)). Inhibition of their degradation to raise the concentrations in synaptic regions using MAOIs restores this function [30,31].

CX157 (3-fluoro-7-(2,2,2-trifluoroethoxy)-phenoxanthiin 10, 10-dioxide) is currently under development for the treatment of major neurological disorders. CX157 causes potent and specific MAO-A inhibition in mammalian brain tissue [32]. MAO-A specifically deaminates serotonin, NE, and tyramine and is inhibited selectively at nanomolar concentrations of clorgyline, while MAO-B is insensitive to inhibition by clorgyline [33]. The first selective inhibitor of MAO-B was L-deprenyl (selegiline) [34] patented as an antidepressant and as psychic energizer [35,36]. Advanced basic studies have suggested the therapeutic utility of selegiline for treatment of both Parkinson’s disease and depression [37]. The evidence of selegiline having neuroprotective action is still followed and proven by the follow-up pro-pargylamino derivative, rasagiline, as shown recently in a study [37,38,39]. Researchers discovered isoxazole derivatives that have good MAO-B inhibition in the micromolar range. They also describe the synthesis and formulation of the molecule for MAO inhibition [28]. 

### Limitations of Currently Approved MAOIs and Current Approach for Development of New MAOIs

MAOIs are powerful antidepressant agents and have been proven to reduce depression symptoms. The most commonly used drugs for neurological disorders are phenelzine, tricyclic antidepressants (TCAs), and benzodiazepines [40,41]. MAOIs are not recommended for use in combination with TCAs. MAOIs in clinical use have different side effects which can interfere with the treatment of depression. The most common adverse effects of MAOIs are nausea, diarrhea, insomnia, drowsiness, dizziness, headaches, and constipation, among others. MAOIs can potentially produce interactions including drug–drug, drug–food, and drug–herbal interactions. Use of antidepressant drugs such as bupropion, paroxetine fluoxetine, nortriptyline, and amitriptyline should be avoided with MAOIs. MAOIs also show interactions with pain medications such as cyclobenzaprine, mirtazapine, meperidine, tramadol, and St. John’s wort. Interactions between MAOIs and foods containing tyramine lead to hypertensive crisis. Hypertension is characterized by severe headache, high blood pressure, sweating, and nausea. To minimize the risk of hypertension effects, patients on MOAIs should take a low-tyramine diet and avoid foods which contain degraded protein such as aged cheeses, sauerkraut, tyramine-rich foods, and smoked meats [42,43]. Other side effects of MAOIs reported are weight gain, impaired sexual functioning, insomnia, and anticholinergic effects (dry mouth and constipation), which occur after long-term treatment with antidepressants [31,44,45,46,47,48]. First-generation MAOIs (non-selective and irreversible) may also cause serious side effects such as hepatotoxicity, orthostatic hypotension, and most importantly, hypertensive crisis that occurs following the consumption of tyramine-rich foods such as aged cheeses [42,49]. Patients using selegiline for Parkinson’s disease in combination with levodopa can suffer with the side effects such as anorexia or nausea, dry mouth, dyskinesia, and orthostatic hypotension [49]. Hydrazine derivatives such as phenelzine used as an antidepressant drug have shown serious adverse effects, for example, liver toxicity, hypertensive crises, hemorrhage, and in some cases, death. Liver toxicity has been reported specifically with hydrazine-derived inhibitors. This led to the development of non-hydrazine MAOIs such as tranylcypromine and pargyline. However, hypertensive crises due to treatments with these MAOIs have continued to cause problems [50]. The greatest disadvantages of TCAs and benzodiazepines are overstimulation, heart palpitations, and sweating. Therefore, these drugs have been discontinued for clinical use [50,51].

Antidepressants acting as reversible inhibitors of MAO-A (also referred as RIMAs) have much less impact on the clinical psychopharmacology than other modern classes of medications, such as selective serotonin reuptake inhibitors (SSRIs). RIMAs are distinguished from the previously used MAOIs by their selectivity and reversibility [32]. As suggested with the use of irreversible MAOIs, dietary restrictions are not required during RIMA therapy and hypertensive calamities are relatively less common. Little evidence has emerged to suggest that RIMAs distribute older MAOIs’ efficacy for treatment of depression characterized by prominent reverse neurodegenerative features [52]. Based on the available evidence, RIMAs appear to have a limited but important role in the differential therapeutics of depressive disorders [21]. 

## 4. Traditional and Psychoactive Medicinal Plants and Herbal Formulations for Treatment of Neurological Disorders

The ancestral traditional medical systems worldwide have widely used medicinal plants for the treatment of different ailments, including neurological disorders. Recent review reports have described the use of psychoactive medicinal plants and herbal formulations for treatment of different neurological disorders [53,54,55,56,57,58]. Among the medicinal plants with neurological effects, a prominent example is *Banisteriopsis caapi*, a woody vine plant that generally grows in the Amazonian basin. *B. caapi* is an ingredient of the hallucinogenic and sacred drink popularly known as ayahuasca “aya”, a Quechua (South American language) word which means “vine of the dead”; “aya” is also known locally as “hoasca, caapi, oasca” (Brazil) and “yage” (Colombia) [59]. For the preparation of the sacred drink in healing purposes or divine exploration, *B.*
*caapi* is utilized as an auxiliary plant along with primarily *Psychotria viridis* (chacruna) or *Diplopterys cabrerana* (oco yage). *B. caapi* contains beta-carbolines, which have therapeutic properties for neurological disorders [60,61,62]. Other notable examples are *Pegnanum harmala* (wild rue), *Rhodiola rosea* (roseroot), and *Crocus sativus* (saffron) for depression; *Passiflora incarnata* (passionflower), *Scutellaria lateriflora* (scullcap), *Gingko biloba* (gingko), and *Zizyphus jujuba* (jujube) for early dementia and anxiety disorders; and *Piper methysticum* (kava-kava) for phobic, panic, and obsessive-compulsive disorders [53,54]. Moreover, many species of mind-altering (psychodysleptic) plants have been used by humans throughout the planet to reach mind distortion states; among those, a few examples have been utilized for therapeutic aims, including *Cannabis sativa* (cannabis), *Tabernanthe iboga* (iboga), *psychotria viridis* (chacruna), and *Papaver somniferum* (opium poppy) [63]. 

MAO-A inhibitors have been demonstrated to be effective antidepressants, and MAO-B inhibitors are used in the treatment of neurodegenerative diseases including PD, as mentioned in Section 3 [35]. The mechanistic aspects of the potential therapeutic applications of MAO-B inhibitors in Alzheimer’s disease were reviewed recently [35]. In this context, several medicinal plants have shown pharmacological effects similar to MAOIs. Recent review reports have summarized the medicinal plants as effective as MAOIs [63,64]. This review presents an extensive survey of the chemical properties of MAO inhibitory constituents identified from natural product sources presented as different chemical classes. This may be useful for further follow-up studies with these MAO inhibitory constituents as new drugs leads for structure activity analysis and further optimization of these leads. Utility of the natural product MAO inhibitory constituents as neuroprotective agents and the therapeutic application of natural products for neuroblastoma are also discussed. 

Electronic searches were specifically conducted on the literature using major databases including PubMed, Google Scholar, SciFinder^®^, and Web of Science. The keywords used in the searches were a combination of the words “natural products’, “alkaloids”, “flavonoids”, “phenols”, “terpenes”, “monoamine oxidase inhibitors”, “MAO”, and “MAOI”. We mostly selected published reports that reported natural products or natural product derivatives with inhibition of MAO-A or-B with IC_50_ < 100 μM. These reports were further grouped into different chemical classes of natural product MAOIs. The natural product constituents tested against MAO-A and/or MAO-B are included in the data tables.

## 5. Different Classes of Natural Product MAOIs

Natural products are an important class of nature-based MAOIs. In the present scenario, several research labs are working to identify and characterize the class of natural products with medicinal value of identified lead compounds. In this review, we summarized different classes of natural product MAOIs. The natural products are generally classified based on chemical classes of secondary metabolites, such as alkaloids, flavonoids, coumarins, xanthones, terpenoids, sterols, and phenolic compounds. Scientists believe that many more compounds of these classes still can be discovered in nature. These natural products and compounds are responsible for a plethora of therapeutic uses, including for treatment of neurological disorders. Various classes of natural product MAOIs are described below. Table 2, Table 3, Table 4, Table 5, Table 6 and Table 7 include selected natural products, the natural source, their MAO-A or MAO-B IC_50_ values, *K*_i_ (when available), selective index (SI; MAO-A or -B), and the source of the enzyme used for MAO inhibition analysis.

### 5.1. Alkaloids

Alkaloids are “*cyclic compounds containing nitrogen in a negative oxidative state*” that are of limited distribution in living organisms [65]. Alkaloids are natural products of non-peptidic origin containing a nitrogen atom, and their basic characteristic is revealed in the name derived from alkaline that means basic. Being one of the most diverse kind of secondary metabolites discovered in living organisms, alkaloids present an array of chemical structures, biosynthetic pathways, and pharmacological activities. While alkaloids have been conventionally isolated from plants, an increasing amount remains to be found in animals, insects, microorganisms (bacteria and fungi), and marine sources (invertebrates and microorganisms) [66]. Several alkaloids have been reported for MAO-A and -B inhibitory activity. Herein, we describe some notable examples of alkaloid types that inhibit MAO enzymes. The chemical structures for the selected alkaloid MAOIs are shown in Figure 4 and Figure 5. Generally, short alkaloids from *Evodia* are moderately selective toward MAO-B. Monoterpene alkaloids, tetra and pentacyclic berberine types, have shown to be selective to MAO-A inhibition, and quinolone and quinoline alkaloids are low- and high-selective MAO-B inhibitors, respectively. Crinine-type alkaloids have been proposed as MAO-A inhibitors; however, no MAO-B inhibition has been reported (Table 2, Figure 4). Harmine-type (beta-carbolines) alkaloids are highly selective toward MAO-A isoenzymes, and tetra and pentacyclic indole alkaloids are also highly potent MAO-A inhibitors; on the contrary, azepine-indole alkaloids are relatively selective toward MAO-B (Table 2, Figure 5).

#### 5.1.1. MAO Inhibitory Activity of Piperine, Quinolone, and Isoquinoline Alkaloids 

Piperine (**1**) alkaloid is the main component in black pepper (*Piper nigrum*) and in general in the *Piper* genus, which has a wide spectrum of pharmacological activities, including MAO inhibition. For example, from *Piper longum,* piperine **1** and methylpiperate **2** were isolated and their MAO inhibition was evaluated using rat brain mitochondrial fraction [67]. The enzyme inhibition kinetics study for piperine **1** showed that the mode of inhibition for MAO-A was mixed type with a *K*_i_ value of 35.8 μM and for MAO-B was competitive with a *K*_i_ value of 79.9 μM [68]. In another study, piperine **1** showed an inhibitory effect against MAO-A with an IC_50_ value of 20.9 μM and for MAO-B with an IC_50_ value of 7.0 μM. The MAO inhibition by piperine **1** was reversible as determined by the recovery of enzyme activity after removal of the inhibitor by dialysis of the incubation mixture [69]. The chemical structure of **1** (Figure 4 and Table 2) consists of a free phenolic –OH group, and it was proposed that the inhibition could probably be initiated by the hydrogen bonding of its amide active protons such as –NH–, –OH, and –SH in the active sites of MAO-A and -B [68]. Methylpiperate **2** was isolated from the fruits of *P. longum* and evaluated for inhibitory effects on the MAO-A activity on mouse brain homogenates yielding an IC_50_ value of 3.6 μM; guineesine **3** showed moderate inhibition of total MAO with an IC_50_ of 139.2 μM [69,70]. Several *N*-amide derivatives of piperine **1** were synthesized and tested for MAO inhibition. Piperic acid *N*-propyl amide **4** inhibited MAO isolated from rat brain mitochondrial homogenate with an IC_50_ value of 45 nM for MAO-B and an IC_50_ value of 3.66 μM for MAO-A [71].

Coptisine **5**, as well as other isoquinoline alkaloids, were isolated from *Coptis japonica*. The isolated alkaloids from *C. japonica* were evaluated for their inhibitory effect on MAO from mouse brain fraction. The result showed an inhibitory effect of **5** on MAO-A with an IC_50_ value of 1.8 μM but none on the MAO-B enzyme [72]. Earlier reports have mentioned that various alkaloids including berberine **6** and palmatine **7** and **5** inhibited MAO enzymes [73,74]. Berberine **6** and palmatine **7** exhibit a non-competitive inhibition of MAO with *K*_i_ values of 44.2 μM and 58.9 μM, respectively [75]. 

Isoquinoline alkaloids including avicine **8**, nitidine **9,** and chelerythrine **10** (Figure 4) isolated from *Zanthoxylum rigidum* showed potent inhibition of human MAO-A with IC_50_ values of 0.41, 1.89, and 0.55 μM, respectively [76]. Limacine **11** and 2′-*N*-chloromethytetrandrine **12**, two benzylisoquinoline dimers isolated from the roots of *Stephania tetrandra*, were found to have moderate inhibitory effects on total MAO with IC_50_ values of 37.7 and 29.2 μM, respectively [77]. 

Quinolone-type alkaloids also have been reported to possess MAO inhibition; in fact, 1-methyl-2-undecyl-4-(1H) quinolone **13** isolated from *Evodia rutaecarpa* competitively inhibited MAO-B with an IC_50_ value of 15.3 µM but did not inhibit MAO-A [78]. *Evodia*
*rutaecarpa* (Rutaceae) is a well-known traditional Chinese medicine with several therapeutic properties, including analgesic, antiemetic, astringent, hemostatic, uterotonic, cardiotonic, and antihypertensive activities [79]. Seven quinolone alkaloids were isolated from the fruits of *E. rutaecarpa* and their inhibitory effects against MAO enzymes were evaluated using mouse brain mitochondrial fraction. The results found that all compounds were more active against MAO-B compared to MAO-A, with 1-methyl-2-nonyl-4 (1H)-quinolone **14** and 1-methyl-2 [(6Z,9Z)-6,9-pentadecadienyl] -4-(1H) quinolone **15** being the most potent inhibitors for MAO-B with IC_50_ values of 2.3 µM and 3.6 µM, respectively [80]. Based on the quinolone base structure, efforts were made to modulate the activity; thus, the synthesis of a series of quinolone derivatives found *N*-(3,4-Dichlorophenyl)-1-methyl-4-oxo-1,4-dihydroquinoline-3 carboxamide **16** as a selective inhibitor of human MAO-B with a selective index (SI) of ~1887 and an IC_50_ value of 5.3 nM [81]. The crinine-type alkaloids crinamine **17** and epibuphanisine **18** (isolated from *Crossyne guttata*) and haemanthamine **19** and haemanthidine **20** (isolated from *Scadoxus puniceus*) showed inhibition of human MAO-B with IC_50_ values of 14.9, 39.2, 112.0, and 17.20 nM, respectively, with no inhibition of human MAO-A [82].

**Figure 4 molecules-27-04297-f004:**
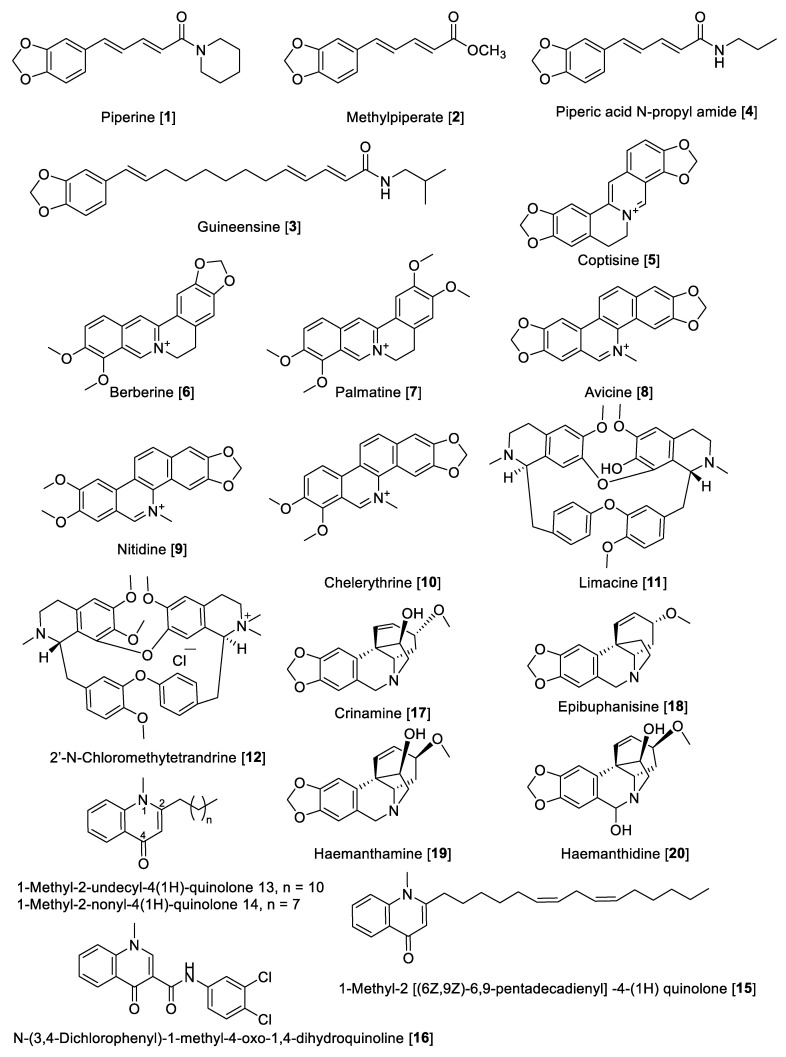
Chemical structure of selected alkaloid MAO inhibitors with piperine, quinolone, and isoquinoline skeletons.

#### 5.1.2. MAO Inhibitory Activity of Beta-Carbolines and Indole-Type Alkaloids

Beta-carbolines derived from indole are endowed with neuropharmacological and neuromodulating properties. Beta-carbolines have been described as potent MAO-A inhibitors [61]. Fernandez de Arriba et al. [83] described the first studies about the kinetic behavior of beta-carboline derivatives as MAO-A inhibitors in the bovine retina. It has also been found that the beta-carbolines harmol **21**, harmine **22**, and harmane **23** are the most potent MAO-A inhibitors, followed by the 3,4-dihydro-beta-carbolines harmalol **24**, harmaline **25**, 1,2,3,4-tetrahydro-beta-carboline (tetrahydro harmine **26**), and norharman **27** [84,85,86]. The MAO inhibitory activity of beta-carbolines present in the extract of *Banisteriopsis caapi*, a component of “Ayahuasca”, and in mixtures with different compounds or plant extracts was evaluated and also compared with other previously reported MAO inhibition data [87,88]. *B. caapi* extracts and two of its components, **22** and **25**, were tested for MAO inhibition using mouse liver homogenate. *B. caapi* extract and harmaline **25** showed a concentration-dependent inhibition for MAO-A with IC_50_ values of 1.24 μg/mL and 4.54 nM, respectively, showing moderate inhibition of MAO-B [88,89]. Phytochemical and biological studies with the specimens of *B. caapi* collected from Hawaii found **22** and **24–26** as the main inhibitors of recombinant human MAO-A with IC_50_ values of 2.0, 2.5, 18, and 74 nM, respectively. Only **22** and **25** were found to be moderately active toward recombinant human MAO-B with IC_50_ values of 20 and 25 μM, respectively [60]. 

Harmane **23** was also isolated from the medicinal plant *Uncaria rhynchophylla* (Cat’s claw) and tested for its inhibitory effect on the total MAO activity of mouse brain homogenate. Harmane **23** was found to be a MAO-A inhibitor with an IC_50_ value of 11.1 µM [90]. Structural and mechanistic studies have demonstrated that harmine **22** shows reversible inhibition and bounds with the active site of the enzyme cavity. Compound **22** interacts with Tyr-69, Asn-181, Phe-208, Val-210, Gln-215, Cys-323, Ile-325, Ile-335, Leu-337, Phe-352, Tyr-407, Tyr-444, and FAD. The seven molecules of water occupy the gap between the inhibitor and these groups. The amide groups of the Gln-215 side chain tightly interact with **22** in human MAO-A or Gln-206 in human MAO-B (Gln-215/206). These results are consistent with previously reported structural analyses of human MAO-A and MAO-B by active site-directed mutagenesis studies [11,12]. The selectivity of the reversible inhibitors is caused by the different size and shape of the substrate and inhibitor cavity regulated by Ile-335 and Phe-208 in MAO-A, which corresponds to Tyr-326 and Ile-199 of MAO-B [91]. 

*Psychotria* (Rubiaceae) is a complex neotropical genus of remarkable interest due to the high content of alkaloids. *Psychotria viridis* is a component of the hallucinogenic beverage known as ayahuasca. The metabolite responsible for its hallucinogenic effects is the alkaloid *N*, *N*-dimethyl tryptamine (DMT) **28**, which is biosynthetically based on the indole scaffold. There are several other alkaloid skeletons produced by the *Psychotria* species, some of them with notable bioactivities [92]. The monoterpene indole alkaloids lyaloside **29**, strictosamide **30,** angustine **31**, vallesiachotamine lactone **32**, *E*-vallesiachotamine **33**, and *Z*-vallesiachotamine **34** (Figure 5 and Table 2) were isolated from *P. laciniata* and were found to be MAO inhibitors with a preference toward human MAO-A with IC_50_ values of 182, 141, 1.10, 0.87, 2.14, and 0.85 μM, respectively, and negligible or low inhibition toward human MAO-B. These alkaloids were also selective inhibitors of butyrylcholinesterase, suggesting that this scaffold can be a multifunctional agent [93]. Recently, the azepine-indole alkaloids cimitrypazepine **35**, fargesine **36**, and nemorosine A **37** were isolated from *Psychotria nemorosa* among other alkaloids and found to inhibit both MAOs but preferentially human MAO-A with IC_50_ values of 1.4, 1.4, and 0.9 μM, respectively [94]. Harmine **22** has been subjected to a clinical trial in combination with DMT **28** to study the network dynamics following the modulation of the serotonin system [95].

Desmodeleganine **38**, a derivative of **28**, was found to inhibit both MAOs at micromolar concentrations. Compound **38** was isolated from the traditional Chinese plant “Sha MaHuang” [96]. Several other alkaloids have also been tested for their inhibitory effect on MAO but have not shown potency of inhibition for MAO-A and -B. Table 2 presents IC_50_ values of selected alkaloid metabolites for inhibition of MAO-A and -B.

**Figure 5 molecules-27-04297-f005:**
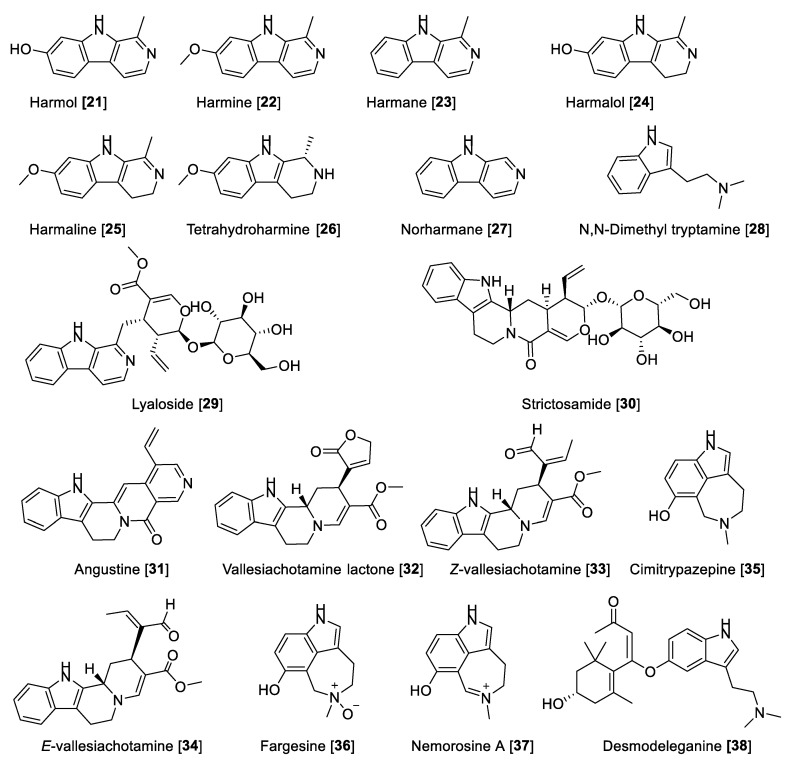
Chemical structure of selected alkaloid MAO inhibitors with indole motif.

**Table 2 molecules-27-04297-t002:** Alkaloid natural product inhibitors of MAO-A and MAO-B.

Compounds	Source	MAO-A		MAO-B		SI	Enzyme Source	References
		IC_50_ (μM)	*K*_i_ (μM)	IC_50_ (μM)	*K*_i_ (μM)	MAOA/B		
Piperine [**1**]	*Piper longum*	49.3	35.8	91.3	79.9	0.54	A	[68]
	*Piper longum*	20.9	19.0	7.0	3.19	2.98	A	[69]
Methylpiperate [**2**]	*Piper longum*	27.1	23.5	1.6	1.3	16.93	A	[70]
piperic acid *N*-propyl amide [**4**]	Piperine derivative	3.66		0.045		81.3	B	[71]
Coptisine [**5**]	*Coptis japonica*	1.8	3.3				A	[72]
Avicine [**8**]	*Zanthoxylum rigidum*	0.41		>100		>0.0041	F	[76]
Nitidine [**9**]	*Zanthoxylum rigidum*	1.89		>300		>0.0063	F	[76]
Chelerythrin [**10**]	*Zanthoxylum rigidum*	0.55		>20		>0.0275	F	[76]
1-Methyl-2-undecyl-4-(1H) quinolone [**13**]	*Evodia rutaecarpa*	338.2		15.3	9.91	22.10	A	[78]
1-Methyl-2-nonyl-4 (1H)-quinolone [**14**]	*Evodia rutaecarpa*	240.2		2.3		104.4	A	[80]
1-Methyl-2 [(6Z,9Z)-6,9-pentadecadienyl] -4-(1H) quinolone [**15**]	*Evodia rutaecarpa*	>400		3.6	3.8	111.1	A	[80]
Quinoline derivative [**16**]	Quinolone derivative	>100		0.0053		>18,867	F	[81]
Crinamine [**17**]	*Crossyne guttata*	0.014					F	[82]
Epibuphanisine [**18**]	*Crossyne guttata*	0.039					F	[82]
Haemanthamine [**19**]	*Scadoxus puniceus*	0.112					F	[82]
Haemanthidine [**20**]	*Scadoxus puniceus*	0.017					F	[82]
Harmol [**21**]	*Banisteriopsis caapi*	0.018					F	[61]
	*Banisteriopsis caapi*	0.5					F	[84]
	*Peganum harmala*	0.352					F	[86]
Harmine [**22**]	*Banisteriopsis caapi*	0.002		20		0.0001	F	[61]
	Diverse vendors	0.06		NR			F	[84]
	*Peganum harmala*	0.008		NR			F	[86]
	*Banisteriopsis caapi*	0.004		>10		>0.0004	A	[89]
Harmane [**23**]	*Banisteriopsis caapi*	0.64		NR			F	[84]
Harmalol [**24**]	*Banisteriopsis caapi*	0.66		NR			F	[84]
	*Peganum harmala*	0.48		NR			F	[86]
Harmaline [**25**]	*Banisteriopsis caapi*	0.002		25		0.00008	F	[61]
	Diverse vendors	0.09		NR			F	[84]
	*Peganum harmala*	0.012		NR			F	[86]
Tetrahydro harmine [**26**]	Diverse vendors	1.52		NR			F	[84]
Norharmane [**27**]	Diverse vendors	4.29		NR			F	[84]
Lyaloside [**29**]	*Psychotria. Laciniata*	182		>100			F	[93]
Strictosamide [**30**]	*Psychotria laciniata*	141		>100			F	[93]
Angustine [**31**]	*Psychotria laciniata*	1.10		138		0.0079	F	[93]
Vallesiachotamine lactone [**32**]	*Psychotria laciniata*	0.87		34		0.025	F	[93]
*E*-vallesiachotamine [**33**]	*Psychotria laciniata*	2.14		120		0.017	F	[93]
*Z*-vallesiachotamine [**34**]	*Psychotria laciniata*	0.85		126		0.0067	F	[93]
Cimitrypazepine [**35**]	*Psychotria nemorosa*	22		1.4		15.71	F	[94]
Fargesine [**36**]	*Psychotria nemorosa*	27		1.4		19.28	F	[94]
Nemorosine A [**37**]	*Psychotria nemorosa*	31		0.9		34.4	F	[94]
Desmodeleganine [**38**]	*Desmonium elegans*	9.33		10.16		0.91	F	[95]

Note: natural products tested on total MAO are not listed. Enzyme source: A = mouse brain crude mitochondrial fraction; B = rat brain mitochondrial MAOs; C = rat liver mitochondrial MAOs; D = mouse liver MAOs; E = human MAO-A and -B over-expressed; F = recombinant human MAO-A and -B.

### 5.2. Flavonoids

Flavonoids are included in the polyphenol family, which are made up of two benzene rings and merged with a short three-carbon chain. These types of compounds are water-soluble polyphenolics containing 15 carbon atoms. One of the carbons of the short chain is always connected to a carbon of one of the benzene rings, either directly or through an oxygen bridge, thereby forming a third middle ring, which can be five or six-membered. Flavones, flavanols, flavanones, anthocyanins, isoflavones, and flavans are the subgroups in flavonoids [97], also including the precursors chalcones, neoflavonoids, and dimers and oligomers. Some flavonoids are responsible for the coloring in the plants, including herbs, fruits, vegetables, etc. The most important dietary sources of flavonoids are green tea, fruits, and vegetables. Green and black tea contain about 25% flavonoids. Fruits such as apple (quercetin) and citrus fruits (rutin and hesperidin) are rich sources of flavonoids [27,98]. Flavonoids are also known for their strong antioxidant properties; it is believed that flavonoids are integrated in a network in plants, activating their defenses to fight stress conditions [99,100]. Flavonoids have also been reported as MAO inhibitors. A series of reports has listed several flavonoids to possess MAO inhibition, including a comprehensive structure activity relationship (SAR) study [27,101,102]. Herein, we describe notable examples of several flavonoids that inhibit MAO, separated by class. Notably, flavones depending on the substituents can be MAO-A or -B inhibitors. Those substituted at the C-4′ position at ring C are mainly MAO-B inhibitors, and those disubstituted at ring C or per-substituted at ring B-turn are relatively MAO-A selective (Table 3, Figure 6). Flavanols are selective MAO-A inhibitors, while isoflavones are reported to be moderately MAO-B selective (Table 3, Figure 6). The chemical structures for the selected flavonoids are shown in Figure 6, Figure 7 and Figure 8. The MAO inhibition profiles including IC_50_ values against MAO-A and MAO-B of these metabolites are presented in Table 3. The sources of MAO employed in these studies are also specified. 

**Figure 6 molecules-27-04297-f006:**
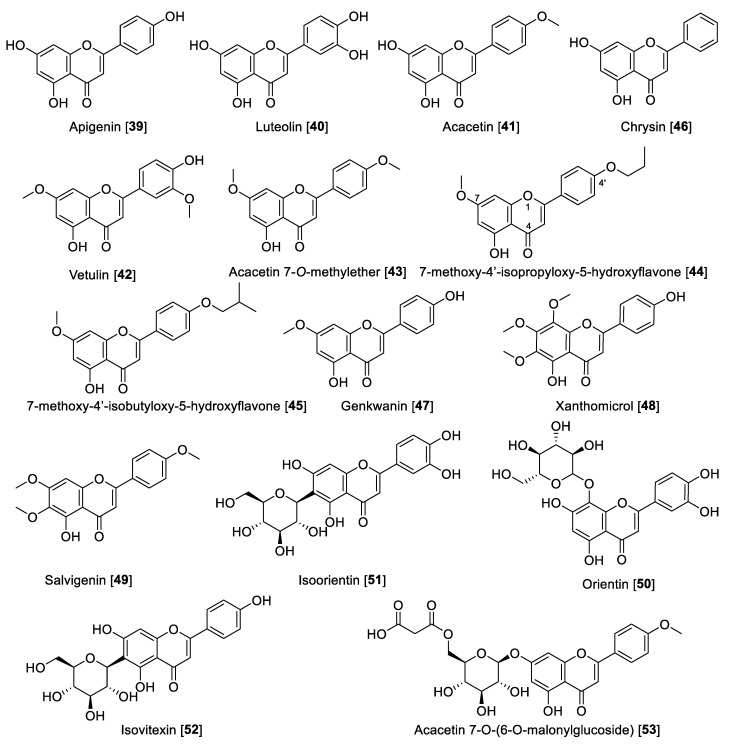
Chemical structure of selected flavone MAO inhibitors.

#### 5.2.1. MAO Inhibitory Activity of Flavones

Apigenin **39**, one of the most notable examples of dietary flavone, is heavily consumed by humans. It is found in many fruits, vegetables, and herbs [103]. Apigenin **39** is also one of the most renowned flavones with several nutritional and pharmacological properties [104]. Regarding its MAO inhibitory activity, **39** isolated from propolis samples was reported to have moderate inhibition toward MAO enzymes, being slightly preferential to human MAO-A than MAO-B with IC_50_ values of 0.64 and 1.12 μM, respectively [105]. Luteolin **40** is another remarkable dietary flavonoid with great pharmacological potential including its use to treat diabetes, Alzheimer’s disease, and depression [106]. Luteolin **40**, isolated from *Cirsium japonicum* var. *maackii* (Compositae) (also known as Korean thistle), was found to be an inhibitor of human MAO-A with an IC_50_ value of 8.57 μM [107]. Acacetin **41**, another interesting dietary flavone, was isolated from the Central American medicinal plant *Calea urticifolia* in a bioassay guided fractionation paradigm as the active MAO inhibitor with IC_50_ values of 121 and 49 nM for human MAO-A and MAO-B, respectively [108]. Later, the studies with medicinal plant *Turnera diffusa* allowed the isolation of the flavones **41**, vetulin **42,** and acacetin 7-*O*-methyl ether **43** (Figure 6 and Table 3) as the active bio-components of MAO inhibition. The study found that **42** preferentially inhibited human MAO-B with an IC_50_ value of 447 nM and an SI of 42. However, most importantly, **43** was found to be a selective and reversible human MAO-B inhibitor with an IC_50_ value of 198 nM and SI of 505 [109]. The search for highly selective reversible MAO-B inhibitors was accomplished based on the computational study of acacetin 7-*O*-methyl ether-scaffold and the active sites of the MAO enzymes. Several derivatives of **43** were designed and found to have 1000- to 3000-fold selectivity toward human MAO-B; for example, 7-methoxy-4′-isopropyloxy-5-hydroxyflavone **44** and 7-methoxy-4′-isobutyloxy-5-hydroxyflavone **45** were found to inhibit human MAO-B with IC_50_ values of 33 and 31 nM, respectively, and an SI of 1921 and 3225, respectively [101]. Several other dietary flavones have been isolated and evaluated by their MAO inhibition. For example, the dietary flavone chrysin **46** isolated from the North African medicinal plant *Cytisus villosus* showed IC_50_ values of 0.25 and 1.04 μM for human MAO-A and MAO-B, respectively [110]. The phytochemical study of *Prunus padus* yielded the flavone genkwanin **47** as a non-selective MAO inhibitor with IC_50_ values of 0.14 for MAO-A and 0.35 μM for MAO-B [111]. Xanthomicrol **48** and salvigenin **49** are two well-known flavones present in the *Sideritis* genus. Xanthomicrol **48** and salvigenin **49** were shown to be human MAO-A inhibitors with *K*_i_ values of 0.76 and 0.54 μM, respectively [112].

Lately, interest has been growing about the health benefits of glycoside flavones. Flavone *O*-glycosides or *C*-glycosides are present in the human diet and are keys to the biosynthesis and functions of plants, several of which have significant reported bioactivities [113]. There are several glycosides reported to display MAO inhibition, for example, orientin **50**, isoorientin **51,** and isovitexin **52** are well-known flavone glycosides isolated from *Vitex grandiflora* (grapes). These flavonoid glycosides were reported to be selective human MAO-B inhibitors with IC_50_ values of 11.04, 11.08, and 21.3 μM, respectively [114]. From *Agastache rugosa* and using a bioguided assay fractionation, **41** and its glycoside acacetin 7-O-(6-O-malonylglucoside) **53** were isolated and found to be MAO inhibitors. Compound **53** was found to be a non-selective MAO inhibitor with IC_50_ values of 2.34 and 1.87 μM toward human MAO-A and MAO-B, respectively [115]. 

#### 5.2.2. MAO Inhibitory Activity of Flavanols, Isoflavones, and Flavanones 

Dietary flavanols are mainly referred to quercetin **54**, kaempferol **55,** and myricetin **56** (Figure 7), are specially found in fruits, vegetables, and beverages, and are also the key components in several plants. Similar to flavones, flavanols are also linked with the antioxidant properties of medicinal plants [116]. Regarding their MAO inhibition property, flavanols have been reported widely. Quercetin **54** isolated from *Hypericum hircinum* showed inhibitory activity for MAO-B with an IC_50_ value of 20 µM and for MAO-A with an IC_50_ value of 10 nM, and the source of the MAO was beef brain mitochondria [117]. Compounds **54** and **55** are common components of wine. A study on the constituents of *Vitis vinifera* and MAO inhibition found that **55** was a potent and selective human MAO-A inhibitor with an IC_50_ value of 0.525 μM and no inhibition of MAO-B [118]. This group also found that **54** is a good MAO-A inhibitor with an IC_50_ value of 3.98 μM and poor inhibition of MAO-B with an IC_50_ of more than 100 μM [118]. The differences in the results of this study [117] and others may be explained by the different sources of the MAO enzymes used for inhibition studies [119]. Myricetin **56** was found to have a moderate inhibitory effect toward human MAO with an IC_50_ value of 9.93 μM [110]. Galangin **57**, a flavonol found in propolis, was found to be a preferential MAO-A inhibitor with an IC_50_ value of 0.13 μM compared to an IC_50_ value of 3.65 μM for human MAO-B [105]. Another interesting flavonol isolated from *Prunus padus* is rhamnocitrin **58**. Rhamnocitrin **58** showed selective reversible inhibition of human MAO-A with an IC_50_ value of 51 nM [120]. 

The isoflavonoid formononetin **59** and the flavanone kushenol F **60** were extracted and isolated from the roots of *Sophora flavescens* and showed significant MAO inhibition in a dose-dependent manner (mouse brain fraction). Formononetin **59** showed IC_50_ values of 21.1 and 11.0 µM for MAO-A and MAO-B, respectively. Kushenol F **60** showed a moderate inhibition of MAO-B with an IC_50_ value of 63.1 µM [121]. Seed extracts of *Psoralea coryfolia* have been used for centuries as chemoprotective and antioxidant agents. The study carried out to find the MAO-inhibiting bioactive components found biochanin A **61** as notable inhibitor. The isoflavone **61** was found to be a selective reversible MAO inhibitor with IC_50_ values of 3.43 and 0.09 μM for human MAO-A and human MAO-B, respectively [122]. The Asian medicinal plant *Maackia amurensis* was reported to be a good source of isoflavonoids. The bioassay guided fractionation of *M. amurensis* bark *extract* yielded isoflavonoids calycosin **62** and 8-*O*-methylretusin **63** as reversible and selective inhibitors of human MAO-B with IC_50_ values of 0.24 and 0.23 μM, respectively [123]. The isoflavone glycitein **64** is one of the major components in *Glycine max* (soybeans) and showed selective inhibition of human MAO-A with an IC_50_ value of 8.30 μM [124]. Naringenin **65,** a common flavanone isolated from *Colvillea racemosae*, was identified as a preferential inhibitor of human MAO-B with an IC_50_ value of 0.272 μM [125].

**Figure 7 molecules-27-04297-f007:**
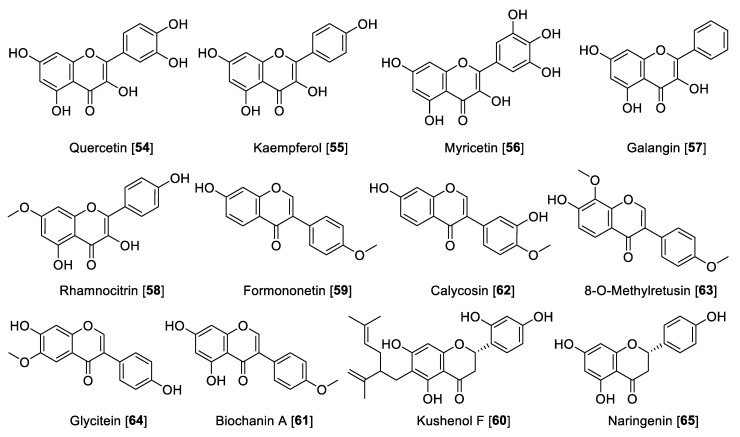
Chemical structure of selected flavanol, isoflavone, and flavanone MAO inhibitors.

#### 5.2.3. MAO Inhibitory Activity of Miscellaneous Flavonoids and Related Compounds

Miscellaneous flavonoids and precursors (Figure 8) such as chalcones have been recently gaining attention for their effects of inhibiting MAO enzymes; for example, chalcones have been proposed as novel and effective MAO-B inhibitors [126,127]. Catechin **66** and epicatechin **67** have been reported to inhibit MAO enzymes. Compounds **66** and **67** isolated from *Uncaria rhynchophylla* and using rat brain homogenate as a source of MAO enzymes showed inhibition of MAO-B with IC_50_ values of 58.9 and 88.6 μM, respectively [128]. The biflavonoids, morelloflavone **68** and GB-2a **69** isolated from *Garcinia gardneriana,* were reported as preferential human MAO-A inhibitors with IC_50_ values of 5.05 and 5.47 μM, respectively [129]. The bichalcone A (3-3″-linked-(2′-hydroxy-4-O-isoprenylchalcone)-(2‴-hydroxy-4″-*O*-isoprenyldihydrochalcone) isolated from the dried bark of *Gentiana lutea* was hydrolyzed to form the product (E)-3-(4′,6-dihyroxy-5′-(3-(3-(2-hydroxyphenyl)-3-oxopropyl)-[1,1′-biphenyl]-3-yl)-1-(2-hydroxyphenyl)prop-2-en-1-one **70**. This hydrolytic product was found to be a MAO inhibitor with an IC_50_ value of 48.7 for MAO-A and 6.2 μM for MAO-B [130]. Isoliquiritigenin **71**, a natural chalcone, showed preferential inhibition of human MAO-B with an IC_50_ value of 0.51 μM and an SI of 44.69 compared to human MAO-A [125]. When the chalcone was saturated and hydroxylated as in the case of colveol A **72,** the selectivity turned toward MAO-A, as denoted by IC_50_ values of 0.62 and 29.9 μM for MAO-A and MAO-B, respectively [125]. Isoliquiritigenin **71**, besides being a potent MAO inhibitor, was also found to have important bioactivity as an antagonist to D_1_ dopamine receptors and agonist to dopamine D_3_ and vasopressin V_1A_ receptors. These finding suggest a potential therapeutic utility of Isoliquiritigenin **71** for neurological disorders [131]. The chalcone 4-hydroxyderricin **73**, isolated from *Angelica keiskei*, was reported to be a selective human MAO-B inhibitor with an IC_50_ value of 3.43 μM and an SI of 1000-fold difference compared to MAO-A [132]. From *Glycine max* Merrill (soybean), an aurone hispidol **74** was isolated. Compound **74** exhibited a preferential inhibition of human MAO-A with IC_50_ values of 0.26 and 2.45 μM for MAO-A and MAO-B, respectively [133]. Medicarpin **75**, a flanovol, was recently reported to have a preferential inhibition of human MAO-B with an IC_50_ value of 0.3 μM [123]. Figure 8 shows the chemical structures and Table 3 presents the IC_50_ values of selected miscellaneous flavonoids.

**Table 3 molecules-27-04297-t003:** Flavonoid natural product inhibitors of MAO-A and MAO-B.

Compounds	Source	MAO-A		MAO-B		SI	Enzyme Source	References
		IC_50_ (μM)	*K*_i_ (μM)	IC_50_ (μM)	*K*_i_ (μM)	MAOA/B		
Apigenin [**39**]	Propolis	0.64	0.125	1.12	0.238	0.57	F	[104]
Luteolin [**40**]	*Cirsium maacki*	8.57		>100		>0.0857	F	[106]
Acacetin [**41**]	*Calea urticifolia*	0.121	0.0592	0.049	0.049	2.46	F	[107]
Vetulin [**42**]	*Turnera diffusa*	18.79		0.447		42.03	F	[108]
Acacetin 7-*O*-methyl ether [**43**]	*Turnera diffusa*	>100		0.198	0.045	<505.05	F	[108]
7-Methoxy-4′-isopropyloxy-5-hydroxyflavone [**44**]	Acacetin derivative	30.74		0.016	0.052	1921.25	F	[101]
7-Methoxy-4′-isobutyloxy-5-hydroxyflavone [**45**]	Acacetin derivative	>100		0.031	0.037	>3225.8	F	[101]
Chrysin [**46**]	*Cytisus villosus*	0.25		1.04	NT	0.24	F	[109]
Genkwanin [**47**]	*Prunus padus*	0.14	0.097	0.35	0.12	0.4	F	[109]
Xanthomicrol [**48**]	*Sideritis spp*		0.76		99.54	0.0076	F	[111]
Salvigenin [**49**]	*Sideritis* spp.		0.54		6.27	0.086	F	[111]
Orientin [**50**]	*Vitex grandiflora*	>100		11.04		>9.05	F	[113]
Isoorientin [**51**]	*Vitex grandiflora*	>100		11.08		>9.02	F	[113]
Isovitexin [**52**]	*Vitex grandiflora*	>100		21.3		>4.69	F	[113]
Acacetin 7-*O*-(6-*O*-malonylglucoside) [**53**]	*Agastache rugosa*	2.34	1.06	1.87	0.38	1.25	F	[114]
Quercetin [**54**]	*Hypericum hircinum*	0.010		20		0.0005	I	[116]
	*Hypericum afrum*	1.52	0.29	28.39		0.053	F	[109]
	*Vitis vinifera*	3.98		>100		>0.039	F	[117]
Kaempferol [**55**]	*Vitis vinifera*	0.525		>100		>0.00525	F	[117]
Myricetin [**56**]	*Hypericum afrum*	9.93	2.24	59.34		0.167	F	[109]
Galangin [**57**]	Propolis	0.13	0.029	3.65	1.998	0.035	F	[104]
Rhamnocitrin [**58**]	*Prunus padus*	0.051	0.097	2.97	0.12	0.017	F	[110]
Formononetin [**59**]	*Sophora flavescens*	21.2		11.0		1.92	A	[119]
	*Maackia amurensis*	4.82		0.19		25.36	F	[121]
Kushenol F [**60**]	*Sophora flavescens*	103.7		63.1		1.64	A	[119]
Biochanin [**61**]	*Psoralea corylifolia*	3.43	0.099	0.09	0.0038	38.11	F	[120]
Calycosin [**62**]	*Maackia amurensis*	70.5		0.24		293.75	F	[122]
8-*O*-Methylretusin [**63**]	*Maackia amurensis*	18.7		0.23		81.30	F	[122]
Glycitein [**64**]	*Pueraria lobata*	8.3		24.9		0.33	F	[123]
Naringenin [**65**]	*Colvillea racemosa*	8.64		0.272		31.76	F	[124]
Catechin [**66**]	*Uncaria rhynchophylla*			88.6	74		B	[127]
Epicatechin [**67**]	*Uncaria rhynchophylla*			58.9	21		B	[127]
Morelloflavone [**68**]	*Garcinia gardneriana*	5.05		66.2		0.076	F	[127]
GB-2a [**69**]	*Garcinia gardneriana*	5.47		56.7		0.20	F	[127]
Bichalcone-derivative [**70**]	*Gentiana lutea*	12.5		6.2	1.2	2.01	B	[127]
Isoliquiritigenin [**71**]	*Colvillea racemosa*	22.66		0.51		44.43	F	[124]
Colveol A [**72**]	*Colvillea racemosa*	0.62		29.90		0.020	F	[124]
4-Hydroxyderricin [**73**]	*Angelica keiskei*	>3000		3.43		>874.63	F	[131]
Hispidol [**74**]	*Glycine max*	0.26	0.10	2.45	0.51	0.10	F	[132]
Medicarpin [**75**]	*Maackia amurensis*	10.2		0.30		34.0	F	[122]

Note: natural products tested on total MAO are not listed. Enzyme source: A = mouse brain crude mitochondrial fraction; B = rat brain mitochondrial MAOs; F = recombinant human MAO-A and -B.

### 5.3. MAO Inhibitory Activity of Coumarins

Coumarins are versatile small lactones, constructed by the combination of a benzene and α pyrone rings fused with each other, which are biosynthetically phenylpropanoid derivatives [134]. Compound **76** was the first example of natural coumarins reported from the *Dipteryx odorata* (*Coumarona odoroata* Syn) beans (tonka beans) way back in 1820. Currently, coumarins are described as important metabolites in natural sources such as medicinal plants and their different parts [135]. The coumarin scaffold shows wide industrial applications and medicinal uses, of which a notable example is warfarin, the most common anticoagulant prescribed worldwide [135]. Coumarins (Figure 9 and Table 4) have been proposed as privilege scaffold and focused on as MAO inhibitors [136,137]. 

Several natural coumarins have been reported as MAO inhibitors, including the monankarins, pigments extracted from the fungi *Monascus anka* [138]. In particular, monankarin A **77** and C **78** were found to exhibit inhibitory activities for MAO in mouse brain homogenate with IC_50_ values of 15.5 and 1.0.7 μM, respectively [138]. From the roots of *Peucedanum japonicum*, a series of coumarins were isolated, and among those, bergapten **79** was found to inhibit total MAO in mouse brain homogenate with an IC_50_ value of 13.8 μM [139]. Additionally, the coumarins aesculetin **80**, aesculetin 7-methyl ether **81**, and scopoletin **82** showed moderate MAO inhibition with IC_50_ vales of 30.1, 32.2, and 45.0 µM, respectively, in mouse brain homogenate. Compounds **80–82** were also isolated from *Artemisia vulgaris,* a well-known European and Asian herb [74]. From the dried flowers of *Hibiscus syriacus*, several coumarins including 8-hydroxy-5,6,7-trimethoxycoumarin **83** and **82** were found to inhibit MAO (brain mouse homogenates) in a dose-dependent manner with IC_50_ values of 44.5 and 19.4 μg/mL, respectively [140]. Geiparvarin **84** and desmethylgeiparvarin **85**, two coumarins isolated from *Geijera parviflora,* were found to be selective toward MAO-B. Geiparvarin **84** showed IC_50_ values of 27 μM and 144 nM for MAO-A and MAO-B, respectively, and desmethylgeiparvarin **85** showed IC_50_ values of 24 μM and 28 nM for MAO-A and MAO-B, respectively [141].

From *Gentiana lutea*, the rearranged coumarin 2-methoxy-3-(1,10-dimethylallyl)-6a,10a-dihydrobenzo(1,2-c)chroman-6-one **86** was isolated and found to be a selective inhibitor of MAO-B with an IC_50_ value of 3.8 μM [130]. Using bioassay guided fractionation of the Asian medicinal plant *Dictamnus albus*, the coumarins 7-(6′*R*-hydroxy-3′,7′-dimethyl-2′*E*, 7′-octadienyloxy) coumarin **87** and auraptene **88** were found to have the highest inhibitory MAO activity. Compound **87** inhibited MAO in a non-selective way with IC_50_ values of 1.3 and 0.5 μM for MAO-A and MAO-B, respectively, while **88** preferentially inhibited MAO-B with an IC_50_ value of 0.6 μM compared to an IC_50_ value of 34.6 μM for MAO-A [142]. Decursin **89**, a tetrahydropyrone coumarin isolated from *Angelica gigas,* was found to be selective toward MAO-A with an IC_50_ value of 1.89 μM compared to an IC_50_ value of 70.5 μM for human MAO-B [143]. From the leaves and twigs of the medicinal plant *Clausena anisum-olens*, anisucoumaramide **90** was isolated. This secondary metabolite **90** showed selective inhibition of human MAO-B with an IC_50_ value of 144 nM and SI of more than 696 compared to MAO-A [144]. From another Apiaceae, *Angelica pubescens,* osthenol **91** was isolated and found to be selective toward human MAO-A with an IC_50_ value of 0.74 μM and SI of >81 compared to MAO-B [145]. The same group also reported that bakuchicin **92** and isopsoralen **93** coumarins isolated from *Psoralea corylifolia* were non-selective MAO inhibitors and that **92** exhibited IC_50_ values of 1.78 and 5.44 μM for human MAO-A and MAO-B, respectively. Compound **93** inhibited human MAO-A and MAO-B with IC_50_ values of 0.88 μM and 2.73 μM, respectively [145]. Umbelliferone **94** and 6-formylumbelliferone **95** isolated from *Angelica decursiva* were found to inhibit MAO enzymes with moderate preference toward human MAO-B with IC_50_ values of 39.16 compared to 147 μM for human MAO-A. The metabolite **95** showed IC_50_ values of 3.23 and 15.31 μM for MAO-A and -B, respectively [145]. The metabolite **95,** besides inhibition of MAO-A, also inhibited lipid peroxidation and Aβ self-aggregation and indicated that 6-formylumbelliferone **95** can be a promising lead for the development of treatments for neurodegenerative diseases [145]. The synthetic studies to modulate the MAO inhibition of coumarins prepared hybrid compounds, making these derivatives more flexible in the case of coumarin-*N*-benzylalkyloxy derivatives [146] and coumarin-chalcones hybrids [147], to mention a few examples. Figure 9 shows the chemical structures for selected coumarins and Table 4 presents the IC_50_ values of selected coumarin compounds.

**Figure 9 molecules-27-04297-f009:**
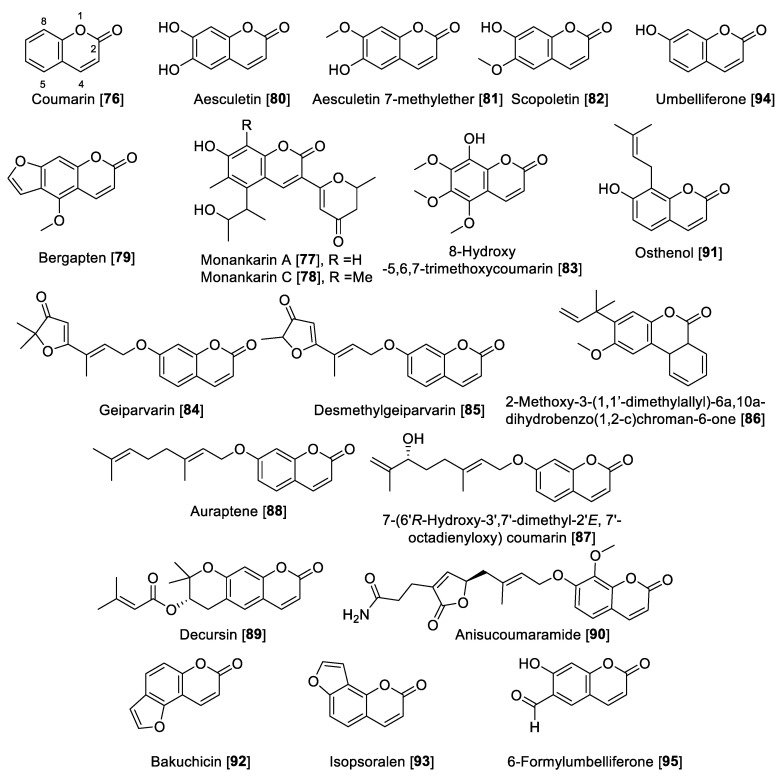
Chemical structure of selected coumarin MAO inhibitors.

**Table 4 molecules-27-04297-t004:** Coumarin natural product inhibitors of MAO-A and MAO-B.

Compounds	Source	MAO-A		MAO-B		SI	Enzyme Source	References
		IC_50_ (μM)	*K*_i_ (μM)	IC_50_ (μM)	*K*_i_ (μM)	MAOA/B		
Geiparvarin [**84**]	*Geijera parviflora*	27		0.144		187.5	B	[142]
Desmethylgeiparvarin [**85**]	*Geijera parviflora*	24		0.028		857.1	B	[142]
2-Methoxy-3-(1,10-dimethylallyl)-6a,10a-dihydrobenzo(1,2-c)chroman-6-one [**86**]	*Gentina lutea*	>100	--	2.9	1.1	>34.5	B	[129]
7-(6′*R*-Hydroxy-3′,7′-dimethyl-2′*E*, 7′-octadienyloxy) coumarin [**87**]	*Dictamnus albus*	1.3	--	0.5	0.46	2.6	A	[141]
Auraptene [**88**]	*Dictamnus albus*	34.6	--	0.6	0.83	57.6	A	[141]
Decursin [**89**]	*Angelica gigas*	0.6		70.5		0.0085	F	[142]
Anisucoumaramide [**90**]	*Clausena anisum-olens*	>100		0.143		699.3	F	[143]
Osthenol [**91**]	*Angelica pubescens*	0.74	0.26	>60	--	>0.012	F	[144]
Bakuchicin [**92**]	*Angelica pubescens*	1.78		5.44		0.32	F	[144]
Isopsoralen [**93**]	*Angelica pubescens*	0.88	0.46	2.73		0.32	F	[144]
Umbelliferone [**94**]	*Angelica decursiva*	39.16		147.37		0.26	F	[145]
6-Formylumbelliferone [**95**]	*Angelica decursiva*	3.23	3.05	15.31	6.81	0.21	F	[145]

Note: natural products tested on total MAO are not listed. Enzyme source: A = mouse brain crude mitochondrial fraction; B = rat brain mitochondrial MAOs; F = recombinant human MAO-A and -B.

### 5.4. MAO Inhibitory Activity of Xanthones, Anthraquinones, and Naphthoquinones

Natural xanthones, anthraquinones, and naphthoquinones are polyphenolic compounds characterized by benzene rings attached to each other by carbonyl and oxygen in the case of xanthones with a C_6_-C_1_-C_6_ skeleton or benzene rings coupled with para-quinones mainly to form naphthoquinone or anthraquinones (C_6_-C_2_-C_6_ sequence), configuring a planar ring structure [148]. The biosynthesis for these secondary metabolites is different depending on the natural source but they can be derived through several pathways. In fact, in plants, anthraquinones are derived mainly by polyketide, mevalonic, and methylerythritol 4-phosphate pathways [149]. On the other hand, in plants, shikimate and fatty acid pathways are predominantly found for xanthones [150]. Recent studies have shown a biosynthetic relationship between xanthones and anthraquinones, especially in bacteria and fungi [151]. Figure 10 and Figure 11 show the structures and Table 5 presents the IC_50_ values of selected xanthones, anthraquinones, and naphthoquinones against MAOs.

Xanthones and xanthone glycosides possess numerous biological and pharmacological potentials. Those compounds have been reported to show several therapeutic properties including antiallergic, anti-inflammatory, antituberculotic, antitumor, and antiplatelet [152,153]. In earlier studies, a series of natural xanthones has been evaluated for their activity to inhibit MAO-A and -B [154,155]. Rat brain mitochondrial extract was the source for MAO. 1-Hydroxy-3,8-dimethoxy-xanthone **96**, 1,3-dihydroxy-7,8-dimethoxy xanthone **97,** and bellidifolin **98** were the potent inhibitors of MAO-A. Xanthones exhibited only moderate inhibition of MAO-B [154,155]. From *Hypericum brasiliense,* a series of compounds was isolated, and among those, 6-deoxyjacareubin **99** and 1,5-dihidroxyxanthone **100** were found to exhibit the best MAO inhibition, using rat brain mitochondria as source of MAO enzymes. Compounds **99** and **100** exhibited slight preferential inhibition of MAO-A with IC_50_ values of 12.0 and 0.73 μM, respectively [156]. 

In a study performed by Gnerre et al., a series of fifty-nine natural and synthetic xanthones was evaluated for inhibition of MAO-A and -B [157]. It was found that the tested xanthones in general were preferential toward MAO-A at low micromolar concentrations. It is important to highlight that the natural xanthone 1,5-dihydroxy-3-methoxy xanthone **101** isolated from *Chironia krebsii*, a medicinal plant growing in the tropical East Africa country Malawi [158], was the most active within the series. Compound **101** exhibited inhibition of MAO-A and MAO-B with IC_50_ values of 40 nM and 33 μM, respectively [157]. Another notable example of natural xanthones is 12b-hydroxy-des-D-garcigerrin A **102**, found in several *Garcinia* species [159], which showed an IC_50_ value of 3.3 μM for MAO-A and no inhibition of MAO-B [157]. Compound **102** also showed antioxidant and antiplasmodial activities [159]. Compound **98** was found to inhibit MAO-A with an IC_50_ value of 0.66 μM [157]. This natural xanthone is frequently isolated from *Gentiana* species [160].

Gentiacaulein **103**, a natural xanthone isolated from *Gentiana kochiana*, was found as one of the two main xanthones present in the plant. Compound **103** exhibited MAO-A inhibition with an IC_50_ value of 0.49 μM [161]. In addition, several other natural xanthones were isolated from *Cudrania tricuspidata* and *Gentianella amarella* and evaluated for their ability to inhibit MAO enzymes. The novel xanthones isolated from *C. tricuspidata* showed poor inhibition of MAO [162], and the only percentage inhibition was reported for the xanthones isolated from *G. amarella* [163]. Dimitrov et al. reported that mangiferin **104**, a glucoside xanthone usually found in the *Hypericum* species, showed selective inhibition of MAO-A with an IC_50_ value of 41 μM and negligible inhibition of MAO-B, using rat liver mitochondria as the MAO source. The authors explained that this MAO inhibition plays a key role in the antidepressant effect of the *Hypericum aucheri* extracts [164].

There are several anthraquinones and naphthoquinones reported to have inhibition toward MAO. Earlier examples are the anthraquinones, norsolorinic acid **105** and averufin **106**, isolated from the fungus *Emericella navahoensis* and which inhibited MAO in mouse liver homogenate. Compound **105** inhibited total MAO with an IC_50_ value of 0.3 μM [165]. Emodin **107**, a common anthraquinone found in plants, exhibited MAO-B inhibition with an IC_50_ value of 35.4 μM [68]. *Lithospermum erythrorhizon* roots are a common component in several preparations used in traditional Chinese medicine (TCM) for treatment of wounds and dermatitis [166]. Bioassay guided fractionation studies of *L. erythrorhizon*-root extracts for MAO inhibition using mouse brain homogenate allowed the isolation of naphthoquinones shikonin **108** and acetyl shikonin **109** as the bioactive components and non-selective MAO inhibitors. Shikonin **108** inhibited MAO-A with an IC_50_ value of 16.4 μM and MAO-B with an IC_50_ value of 13.6 μM; **109** had IC_50_ values of 16.9 and 10.1 μM for MAO-A and MAO-B, respectively [167]. 

Another notable naphthoquinone, 2,3,6-trimethyl-1,4-naphthoquinone **110,** was extracted and isolated from cured tobacco leaves (*Nicotiana tabacum*) and was found to be a competitive inhibitor of human MAO (human liver mitochondrial preparation) with *K*_i_ values of 3 and 6 μM for MAO-A and MAO-B, respectively [168]. Recent studies found using a human MAO assay that **110** inhibited MAOs with IC_50_ values of 1.14 and 7.14 μM for MAO-A and MAO-B, respectively [169]. The same authors evaluated the anxiolytic effects of **110** on a zebrafish model, describing that the compound induced anxiolytic and anxiogenic-like effects on the model tested [169]. Menadione **111**, a naphthoquinone formed by metabolism from the dietary vitamin K (menaquinones), was reported as selective MAO-B inhibitor with a *K*_i_ value of 0.4 μM [170]. Mostert et al. in a study with a series of synthetic and natural naphthoquinones reported juglone **112** and plumbagin **113** to inhibit MAO in a non-selective manner. Compound **112** exhibited IC_50_ values of 1.71 and 4.36 μM for human MAO-A and MAO-B, respectively; plumbagin **113** showed IC_50_ values of 4.91 and 1.09 μM for human MAO-A and MAO-B, respectively [171]. Two natural dyes anthraquinones purpurin **114** and alizarin red **115** were reported to inhibit MAO, and both anthraquinones showed a selectivity toward MAO-A. Compound **114** exhibited an IC_50_ value of 2.50 μM and for **115**, an IC_50_ value of 30.1 μM [172]. In a phytochemical study of the components of *Cassia obtusifolia* seeds, a series of anthraquinones and related compounds were isolated and evaluated for the ability to inhibit human recombinant MAO. The compounds showed selective inhibition of MAO-A, selective and preferential, and notable examples are questin **116** that inhibited human MAO-A with an IC_50_ value of 0.17 μM, as well as aloe emodin **117**, alaternin **118**, rubrofusarin **119**, and toralactone-9-O-β-gentiobioside **120** which were found to be strong inhibitors with IC_50_ values of 2.47, 5.35, 5.9, and 7.3 μM, respectively [173]. 

**Table 5 molecules-27-04297-t005:** Xanthone, anthraquinone, and naphthoquinone natural product inhibitors of MAO-A and MAO-B.

Compounds	Source	MAO-A		MAO-B		SI	Enzyme Source	References
		IC_50_ (μM)	*K*_i_ (μM)	IC_50_ (μM)	*K*_i_ (μM)	MAOA/B		
Bellidolin [**98**]	*Gentiana lactea*.	0.66		>100		>0.0066	C	[157]
6-Deoxyjacareubin [**99**]	*Hypericum brasiliense*	12.0		47.3		0.25	B	[156]
1,5-Dihidroxyxanthone [**100**]	*Hypericum brasiliense*	0.73		76.3		0.0095	B	[156]
1,5-dihydroxy-3-methoxy xanthone [**101**]	*Chironia krebsii*	0.04		33.0		0.0012	C	[157]
12b-hydroxy-des-D-garcigerrin A [**102**]	*Garcinia gerrardii*	3.3		>100		0.033	C	[157]
Gentiacaulein [**103**]	*Gentiana kochiana*	0.22		96.0		0.0022	B	[161]
Mangiferin [**104**]	*Hypericum aucheri*	410		>1000		0.41	C	[170]
Emodin [**107**]	Polygonaceae Fam.			35.4	15.1		B	[68]
Shikonin [**108**]	*Lithospermum erythrorhizon*	16.4	12.8	13.6	13.0	1.20	A	[167]
Acetyl shikonin [**109**]	*Lithospermum erythrorhizon*	16.9	10.5	10.1	6.3	1.67	A	[167]
2,3,6-Trimethyl-1,4-naphthoquinone [**110**]	*Nicotiana tabacum*	3.0		6.0		0.5	E	[168]
		1.14		7.14		1.59	F	[169]
Menadione [**111**]	Vit K derivative		26.0		0.4	65	F	[169]
		10.2		3.02		3.37	F	[170]
Juglone [**112**]	*Juglans* spp.	1.71		4.36		0.39	F	[170]
Plumbagin [**113**]	*Plumbago* spp.	4.91		1.09		4.50	F	[170]
Purpurin [**114**]	*Rubia tinctorum*	2.50	0.422	>40		>0.062	F	[172]
Alizarin red [**115**]	*Rubia tinctorum*	30.1		>60		>0.50	F	[173]
Questin [**116**]	*Cassia obtusifolia*	0.17	4.14	10.58		0.016	F	[173]
Aloe emodin [**117**]	*Cassia obtusifolia*	2.47	0.50	>400		>0.0061	F	[173]
Alaternin [**118**]	*Cassia obtusifolia*	5.35	3.97	4.55		1.17	F	[173]
Rubrofusarin [**119**]	*Cassia obtusifolia*	5.90	4.38	91.40		0.064	F	[173]
Toralactone-9-O-β-gentiobioside [**120**]	*Cassia obtusifolia*	**7.36**	**4.30**	>400		>0.0184	F	[173]

Note: natural products tested on total MAO are not listed. Enzyme source: A = mouse brain crude mitochondrial fraction; B = rat brain mitochondrial MAOs; C = rat liver mitochondrial MAOs; F = recombinant human MAO-A and -B.

### 5.5. MAO Inhibitory Phenols and Polyphenolic Compounds 

The polyphenolic compounds are simple phenolics and their acids and esters produced predominantly by the shikimate pathway include phenylpropanoids and lignans [174]. Several natural phenols were evaluated on MAO rat brain mitochondrial fraction and only Paeonol **121** showed a moderate inhibition toward MAO-A with a *K***_i_** value of 51.1 μM [68]. Ferulic acid **122** showed to be preferential toward MAO-A with an IC_50_ value of 7.55 μM for human MAO-A and 24.0 μM for MAO-B. Gallic acid **123** exhibited IC_50_ of 9.49 μM for human MAO-A; *trans*-cinnamic acid **124,** a non-selective MAO inhibitor, exhibited IC_50_ values of 6.47 and 1.21 μM for MAO-A and -B, respectively; ellagic acid **125** was a selective MAO-B inhibitor with an IC_50_ value of 0.40 μM; and caffeic acid **126** was shown to be a non-selective MAO inhibitor with IC_50_ values of 11.72 and 22.88 μM for human MAO-A and MAO-B, respectively [175]. 

A series of chlorogenic acids was isolated from the flowers of *Lonicera macranthoides*, and from the isolated compounds, only 3,5-di-*O*-caffeoylquinic acid **127** exhibited moderate inhibition with an IC_50_ value of 20.04 μM for MAO-B and no inhibition for MAO-A [176]. From *Lonicera japonica*, using a magnetic nanoparticle MAO-B immobilized affinity solid-phase extraction method, two new chlorogenic acids isochlorogenic A **128** and C **129** were identified along with **122** as the bioactive component in *L. japonica* [177]. Isochlorogenic acid A **128** and C **129** were found to be mixed-type inhibitors for MAO-B with IC_50_ values of 29.0 and 29.77 μM, respectively [177]. Turmeric is a commonly used spice prepared from the rhizomes of *Curcuma longa*. Several bioactivities are attributed to turmeric, and those responsible for that bioactivity are the curcuminoid-type compounds, and in particular, curcumin **130** [178]. The inhibitory effects of **130** and tetrahydrocurcumin **131** on MAO-B were evaluated in a Parkinson’s disease rodent model induced by 1-methyl-4-phenyl-1,2,3,6-tetrahydropyridine. Depletion of dopamine and 3,4-dihydroxy phenyl acetic acid occurs with increased MAO-B activity. Use of **130** and **131** reversed the decrease in dopamine and 3,4-dihydroxy phenyl acetic acid induced by the model [179]. Recently, the ability of **130**, demethoxycurcumin **132,** and bisdemethoxycurcumin **133** to inhibit MAO was reported, where the three compounds inhibited MAO moderately in a non-selective manner. Compound **130** exhibited IC_50_ values of 3.64 and 3.36 μM for human MAO-A and MAO-B, respectively; **131** exhibited IC_50_ values of 3.09 and 2.59 μM, respectively; and **132** exhibited IC_50_ values of 3.24 and 2.45 μM for MAO-A and MAO-B, respectively [180]. Paleacenins A **134** and B **135** are two novel acylphloroglucinols isolated from *Elaphoglossum paleaceum* rhizome, and these two undescribed compounds exhibited high MAO inhibition [181]. Compounds **134** and **135** showed to be a non-selective MAO inhibitor in rat brain mitochondria, and **134** exhibited IC_50_ values of 31.0 and 4.7 μM for MAO-A and-B, respectively. Meanwhile, **135** showed IC_50_ values of 1.3 and 4.4 μM for MAO-A and -B, respectively. Besides their MAO inhibition, **134** and **135** also showed activity toward several cancer cell lines including the prostate, cervix, breast, and colon, denoting a multitarget effect for paleacinins [181]. Figure 12 shows the selected additional polyphenolic compound MAO inhibitors and Table 6 presents the IC_50_ values of selected phenol and polyphenolic compounds.

**Table 6 molecules-27-04297-t006:** Phenol and polyphenolic natural product inhibitors of MAO-A and MAO-B.

Compounds	Source	MAO-A		MAO-B		SI	Enzyme Source	References
		IC_50_ (μM)	*K*_i_ (μM)	IC_50_ (μM)	*K*_i_ (μM)	MAOA/B		
Paeonol [**121**]	*Paeonia* spp.	54.6	51.1	42.5		1.28	B	[68]
Ferulic acid [**122**]			7.55		24.0	0.31	F	[175]
Gallic acid [**123**]		9.49		NR			F	[175]
*t*-Cinnamic acid [**124**]		6.47		1.21		5.34	A	[175]
Ellagic acid [**125**]				0.40			B	[175]
Caffeic acid [**126**]		11.7		22.9		0.51	F	[175]
3,5-di-*O*-caffeoylquinic acid [**127**]	*Lonicera macranthoides*			20.04			B	[176]
Isochlorogenic acid A [**128**]	*Lonicera japonica*			29.05	9.55		J	[177]
Isochlorogenic acid C [**129**]	*Lonicera japonica*			29.77	9.53		J	[177]
Curcumin [**130**]	*Curcuma longa*	3.64		3.36		1.08	F	[180]
Demethoxycurcumin [**132**]	*Curcuma longa*	3.09	0.91	2.59	0.86	1.19	F	[180]
bis-Demethoxycurcumin [**133**]	*Curcuma longa*	3.24	1.40	2.45	0.80	1.32	F	[180]
Paleacenins A [**134**]	*Elaphoglossum paleaceum*	31.0		4.7		6.59	B	[181]
Paleacenins C [**135**]	*Elaphoglossum paleaceum*	1.3		4.4		0.29	B	[181]

Note: natural products tested on total MAO are not listed. Enzyme source: A = mouse brain crude mitochondrial fraction; B = rat brain mitochondrial MAOs; F = recombinant human MAO-A and -B; J = liver porcine.

### 5.6. MAO Inhibitory Terpenes and Terpenoids

Terpenes or terpenoids are one of the broad families of natural products with uncountable pharmacological and biological uses. These secondary metabolites are built based on isoprene units (five carbon building blocks) [182]. There are few terpenoids that have been reported to exhibit MAO inhibition. In a study exploring the use of HPLC-based bioguided activity profiling, *Salvia miltiorrhiza* was screened. The extract of *S. miltiorrhiza* showed a high inhibitory effect on rat liver monoamine oxidase fraction with a preference toward MAO A. From the active fractions based on the HPLC-bioguided assay, the diterpenes dihydrotanshinone I **136**, cryptotanshinone **137**, and tanshinone I **138** were identified as the bioactive compounds with IC_50_ values of 23, 80, and 84 μM, respectively [183]. Recently, a triterpene saponin asiaticoside D **139**, isolated from *Centella asiatica* (a herbal product used in Ayurvedic medicine) following a bioassay guided fractionation, exhibited MAO inhibition [184]. Compound **139** was found to be a non-selective inhibitor with IC_50_ values of 4.0 and 1.3 μg/mL for MAO-A and MAO-B, respectively [184]. Illudinine **140**, a sesquiterpene-alkaloid with an illudalane skeleton, was found to be an inhibitor of human MAO-B with an IC_50_ value of 18.3 μM. Several synthetic derivatives of illudinine have been synthesized; however, none of these compounds were found to be superior to illudinine [185]. In a bioassay guided fractionation study of the *Zingiber officinale* rhizomes, several monoterpenes showed potential MAO-A inhibitory properties [186]. The monoterpenes geraniol **141** and (-) terpinen-4-ol **142** were found to be inhibitors of MAO-A, but only a percentage of inhibition was reported [186]. Figure 13 shows the chemical structures for selected terpenoid MAO inhibitors and Table 7 presents the IC_50_ values of selected terpenoid compounds.

### 5.7. MAO Inhibitors from Marine Sources

In the last three decades, the study of marine environments has been the source of great chemical diversity, and at least six isolated compounds from marine organisms have been FDA approved, with several others currently in clinical trials [187,188]. Several marine natural products have been reported with inhibitory activity toward MAO enzymes, including aplysinopsins, piloquinones, anthiactins, bromopyrroles, caulerpins, and astaxanthins, which have been compiled in a recent review [189]. As an update of the review mentioned before, two well-known phlorotannins isolated from the edible brown alga *Eisenia bicyclis* were found to be MAO inhibitors [190]. The two phlorotannins named eckol **143** and dieckol **144** showed a preferential human MAO-A inhibition with IC_50_ values of 7.2 and 11.43 μM, respectively [190]. Phlorofucofuroeckol-A **145** (PFF-A), another phlorotannin isolated from *Ecklonia stolonifera* (edible brown alga), showed non-selective MAO inhibition. PFF-A **145** showed IC_50_ values of 9.22 and 4.89 μM for human MAO-A and MAO-B, respectively [191]. Compounds **144** and **145** also exhibited dopamine D_3_R and D_4_R agonism as well as D_1_, serotonin 5-HT_1A_, and neurokin NK_1_ antagonism [191]. From the red alga *Symphyocladia latiuscula,* a series of bromophenols was isolated and 2,3,6-tribromo-4,5-dihydroxybenzyl methyl ether **146** and bis-(2,3,6-tribromo-4,5-dihydroxybenzyl) ether **147** exhibited moderate inhibition of human MAO-A with IC_50_ values of 63.2 and 89.3 μM, respectively [192]. Two chromenones isolated from the marine microorganism *Streptomyces* sp (CNQ-031), 5,7-dihydroxy-2-isopropyl-4H-chromen-4-one **148** and 5,7-dihydroxy-2-(1-methylpropyl)-4H-chromen-4-one **149**, were found to be inhibitors for human MAO. Compound **148** inhibited human MAO-A with a 10-fold difference than for MAO-B with IC_50_ values of 2.7 and 27.0 μM, respectively, while **149** showed a non-selective inhibition with IC_50_ values of 6.92 and 3.42 μM for human MAO-A and MAO-B, respectively [123]. Figure 14 shows marine compounds with MAO inhibition and Table 7 presents the IC_50_ values of marine MAO inhibitors.

### 5.8. MAO Inhibitors from Miscellaneous Classes of Natural Products

Small polyketides have been reported as MAOIs; for example, desmethoxyyangonin **150** isolated from the medicinal plant *Renealmia Alpinia* was found to be a MAO inhibitor, being selective toward human MAO-B with an IC_50_ value of 0.12 μM compared to an IC_50_ value of 1.85 μM for MAO-A [18]. Desmethoxyyangonin **150** belongs to the well-known kavalactone-type secondary metabolites which are the major constituents isolated predominantly from *Piper methysticum* roots, the principal ingredient of the kava drink [193]. The kava drink has been used for centuries by the natives of the pacific islands of Vanuatu, Fiji, Tonga, Samoa, and Micronesia due to its psychoactive and medicinal properties. The kava drink is also a popular drink in Western countries [194]. Besides **150**, the kavalactones: (+)-kavain **151**, (+)-7,8-dihydrokavain **152**, (+)-methysticin **153**, (+)-7,8-dihydromethysticin **154**, and yangonin **155** were reported to be selective MAO-B inhibitors with IC_50_ values of 4.34, 8.23, 0.42, 0.85, and 0.085 μM, respectively. Only **153** and **155** exhibited inhibition of human MAO-A with IC_50_ values of 8.12 and 1.29 μM, respectively [195]. Another small polyketide, (*S*)-5-methymellein **156**, a small lactone isolated from the endogenous fungus *Rosellinia corticium* within the lichen *Pseudevernia furfuracea,* exhibited MAO inhibition. Compound **156** was identified through bioassay guided fractionation and exhibited IC_50_ values of 5.31 and 9.15 μM for human MAO-A and MAO-B, respectively [196]. Its enantiomer, (*R*)-5-methylmellein **157**, was also isolated from the fermented mycelia of *Xylaria nigripes,* and **157** showed moderate selectivity toward MAO-A with IC_50_ values of 4.6 and 38.5 μM for human MAO-A and MAO-B, respectively [197]. The enantiomers were subjected to a synthetic study aiming to find better inhibitory agents, and the study found a selective human MAO-B inhibitor in a pyrimidyl derivative, (*R*)-3-ethyl-8-hydroxy-5-methyl-7-(pyrimidin-5-yl)-3,4-dihydronaphthalen-1(2H)-one **158**, with an IC_50_ value of 0.06 μM for human MAO-B and SI value of >830 [197]. From another endogenous fungus *Diaporthe mahothocarpus* isolated from the lichen *Cladonia symphycarpia*, three polyketides were isolated with potent MAO inhibition. The polyketides alternariol **159**, 5′-hydroxy-alternariol **160**, and mycoepoxydiene **161** were isolated following a bioassay guided paradigm and were found to be MAO-A inhibitors. Compound **159** exhibited the best value with an IC_50_ of 0.020 μM, while **160** and **161** exhibited IC_50_ values of 0.31 and 8.7 μM for human MAO-A, respectively [198]. The in vitro evaluation of several endogenous endocannabinoids found that virodhamine **162** inhibited both MAO-A and -B, being preferential to MAO-B with IC_50_ values of 38.70 and 0.71 μM for human MAO-A and MAO-B, respectively [199]. Figure 15 shows the structures and Table 7 presents the IC_50_ values of selected miscellaneous natural product classes on natural product inhibitors.

**Table 7 molecules-27-04297-t007:** Terpenoids, marine sources, and miscellaneous natural product inhibitors of MAO-A and MAO-B.

Compounds	Source	MAO-A		MAO-B		SI	Enzyme Source	References
		IC_50_ (μM)	*K*_i_ (μM)	IC_50_ (μM)	*K*_i_ (μM)	MAOA/B		
Dihydrotanshinone I [**136**]	*Salvia miltiorrhiza*	23					F	[183]
Cryptotanshinone [**137**]	*Salvia miltiorrhiza*	80					F	[183]
Tanshinone I [**138**]	*Salvia miltiorrhiza*	84					F	[183]
Illudinine [**140**]	*Clitocybe illudens*			18.3			F	[185]
Eckol [**143**]	*Eisenia bicyclis*	7.20	20.26	83.44	162.8	0.086	F	[185]
Dieckol [**144**]	*Eisenia bicyclis*	11.43	20.28	43.42	18.50	0.26	F	[190]
Phlorofucofuroeckol-A [**145**]	*Ecklonia stolonifera*	9.22	5.18	4.89	2.69	1.88	F	[190]
2,3,6-Tribromo-4,5-dihydroxybenzyl methyl ether [**146**]	*Symphyocladia latiuscula*	63.16	25.4	105.13	40.7	0.60	F	[191]
*bis*-(2,3,6-Tribromo-4,5-dihydroxybenzyl) ether [**147**]	*Symphyocladia latiuscula*	89.31	22.8	102.53	35.5	0.87	F	[191]
5,7-Dihydroxy-2-isopropyl-4H-chromen-4-one [**148**]	*Streptomyces* spp.	2.70		27.0		0.10	F	[192]
5,7-Dihydroxy-2-(1-methylpropyl)-4H-chromen-4-one [**149**]	*Streptomyces* spp.	6.92		3.42		2.02	F	[192]
Desmethoxyyangonin [**150**]	*Renealmia alpinia*	1.85	0.922	0.12	0.031	15.41	F	[22]
(+)-Kavain [**151**]	*Piper methysticum*	19.0		5.34		3.55	F	[193]
(+)-7,8-Dihydrokavain [**152**]	*Piper methysticum*	>100		8.23		12.15	F	[193]
(+)-Methysticin [**153**]	*Piper methysticum*	8.12		0.42		19.33	F	[193]
(+)-7,8-Dihydromethysticin [**154**]	*Piper methysticum*	23.2		0.855		27.13	F	[193]
Yangonin [**155**]	*Piper methysticum*	1.29	1.12	0.085	0.226	15.17	F	[193]
(*S*)-5-Methymellein [**156**]	*Rosellinia corticium*	5.31	2.45	9.15		0.58	F	[194]
(*R*)-5-Methylmellein [**157**]	*Xylaria nigripes*	4.6		38.5		0.11	F	[195]
(*R*)-3-Ethyl-8-hydroxy-5-methyl-7-(pyrimidin-5-yl)-3,4- dihydronaphthalen-1(2H)-one [**158**]	Methylmellein derivative	0.06		>50		>0.012	F	[195]
Alternariol [**159**]	*Diaporthe mahothocarpus*	0.020	0.0075	20.7		0.00096	F	[196]
5′-Hydroxy-alternariol [**160**]	*Diaporthe mahothocarpus*	0.31	0.116	>40		0.00775	F	[196]
Mycoepoxydiene [**161**]	*Diaporthe mahothocarpus*	8.7	3.76	>40		>0.21	F	[196]
Virodhamine [**162**]		38.7		0.71	0.258	54.50	F	[196]

Note: natural products tested on total MAO are not listed. Enzyme source: F = recombinant human MAO-A and -B.

## 6. Natural MAO Inhibitors in Neuroblastoma

Therapeutic applications of MAOIs have been established primarily for the treatment of depression and other neurological disorders. Oxidative deamination of monoamines by MAO produces oxidative stress [200]. Recent studies have suggested the role of MAO-A in antitumor immunity [201]. An enhanced T cell immunity and suppression of tumor growth were reported in MAO-A knockout mice. Treatment with clinically approved MAO-A inhibitors such as phenelzine, moclobemide, and clorgyline produced tumor suppression in preclinical animal models [201]. The role of MAOs in tumor progression and metastasis suggests these enzymes as potential anticancer drug targets [202]. Neuroblastoma, known to originate from the undifferentiated neural crest cells which differentiate into malignant neuroblastoma, is the most common extracranial solid tumor in children. Excess production of catecholamines in neuroblastoma tumor cells has been reported [203]. This may be the cause for severe hypertension associated with neuroblastoma [204,205,206]. Elevations in the levels of urinary metabolites of catecholamines vanillylmandelic acid and homovanillic acid are routinely assessed as prognostic biomarkers of neuroblastoma [207,208]. A report from the children oncology group suggested a correlation between the elevated expression of the vesicular monoamine transporter with clinical features, tumor biology, and metaiodobenzylguanidine (MIBG) avidity in neuroblastoma [84]. MOA-A has been shown to promote protective autophagy in human SH-SY5Y neuroblastoma cells through Bcl-2 phosphorylation [209]. These studies suggest the potential utility of MAO-A inhibitors for treatment of neuroblastoma. Isatin, an endogenous MAO inhibitor, caused a dose-dependent switch from apoptosis to necrosis in human neuroblastoma cells [210]. The natural product MAO inhibitors may also have potential utility for treatment of neuroblastoma tumors. Harmine **22**, the potent natural product alkaloid inhibitor of MAO-A, has been shown to induce apoptosis and inhibit cell proliferation of several human cancer cell lines [211,212]. Harmine **22** also induced apoptosis in different neuroblastoma cell lines such as SKNBE and KELLY (MYCN amplified) and SKNAS and SKNFI (MYCN non-amplified) [213]. Induction of progressive apoptosis of human neuroblastoma cells by harmine was also attributed to the activation of caspase-3/7 and caspase-9 [214]. Selective inhibition of dual-specificity tyrosine phosphorylation-regulated kinase (DYRK) family proteins and mitogen-activated protein kinase by harmine [214] led to the investigation of the expression of DYRK family kinases in neuroblastoma tumors [213]. A clinical study suggested the role of DYRK2 in tumorigenesis of neuroblastoma. Inhibition of DYRK2 by **22** [212,214] and activation of caspase-mediated apoptosis in neuroblastoma cells [213] suggest the treatment of neuroblastoma by **22**. Norharman **27**, a potential MAO-A inhibitor, also induced apoptotic cell death in human neuroblastoma cells [215]. Berberine treatment produced the induction of neuronal cell differentiation, attenuated cancer stemness markers, and potentiated G_0_/G_1_ cell cycle arrest in neuro2a (N2a) neuroblastoma cells [216]. Concurrent inhibition of β-adrenergic signaling pathways with a β-blocker and inhibition of MAO-A with berberine **6** produced attenuation of tumor growth and an increase in differentiation of cells [217]. 

## 7. Concluding Remarks

This review presents a comprehensive survey of natural products, predominantly isolated or purified from plant sources, which have shown promising MAO inhibitory activity. The natural product MAOIs may be utilized as potential leads for new drug discovery and safer alternatives for treatment of neurological disorders. The neuroactive effects of several traditionally used herbal formulations have been attributed to the presence of the constituents with selective inhibition of MAOs; for example, the potent inhibition of MAO-A by harmala alkaloids is responsible for the hallucinogenic effects of ayahuasca. The survey was further extended to analyze different classes of natural product MAOIs and selectivity of the inhibition by these constituents. Natural products with selective and reversible inhibition of MAOs may be advanced to clinical evaluation for treatment of disorders linked to pathophysiological consequences of the metabolism of biogenic monoamines. 

Selective MAO-A and -B inhibitors isolated from natural sources may be safer alternatives for the treatment of neurological disorders. A greater understanding of the natural product inhibitors for MAOs may improve the drug discovery and treatment of neurological disorders. Evaluation of MAOIs as constituents of natural product ingredients used in dietary supplements and traditional herbal preparations may also be important for preventing adverse drug–dietary supplement interactions. 

Naturally occurring MAOIs are good alternatives for the treatment of depression, anxiety, Parkinson’s disease, and neurodegenerative diseases. Non-selective and irreversible classical MAOIs are characterized by the risk of inducing hypertensive crisis when foods rich in tyramine are ingested. However, selective MAO-B inhibitors do not show such interactions. Several natural products with selective reversible inhibition of MAO-B have been identified. The selective MAO-B inhibitors and RIMAs are potential and safer alternatives to irreversible MAOIs. A greater understanding of the pharmacodynamic and pharmacokinetic properties of the natural product MAOIs may improve treatment of patients in the future. Another challenge for future basic and clinical research is to examine the potential preventive medications that slow the physiological and other age-related activities. The natural products are also an important source of MAOIs that can slow the age-related behavioral problems and decrease susceptibility to senile depression, Parkinson’s disease, and Alzheimer’s disease. Characterization of MAO inhibitory constituents of natural products traditionally used as psychoactive preparations or for treatment of neurological disorders may help in understanding the mechanisms of action and in optimization of these preparations for desired bioactive properties. Potential therapeutic applications of natural product MAOIs for the treatment of neuroblastoma are also discussed.

## Figures and Tables

**Figure 2 molecules-27-04297-f002:**
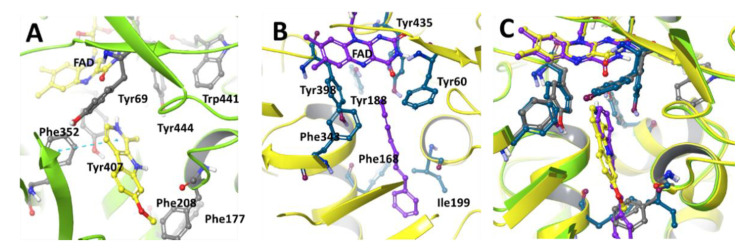
Active site configuration of human (**A**) MAO-A (PDB structure 2Z5X) with harmine and (**B**) MAO-B (PDB structure 1OJ9) with 1,4,-diphenyl butene, depicting key amino acids lining the enzyme active site. (**C**) Overlaid view of MAO-A and MAO-B substrate binding site. Harmine and 1,4,-diphenyl butene are selective reversible inhibitors of MAO-A and MAO-B, respectively.

**Figure 3 molecules-27-04297-f003:**
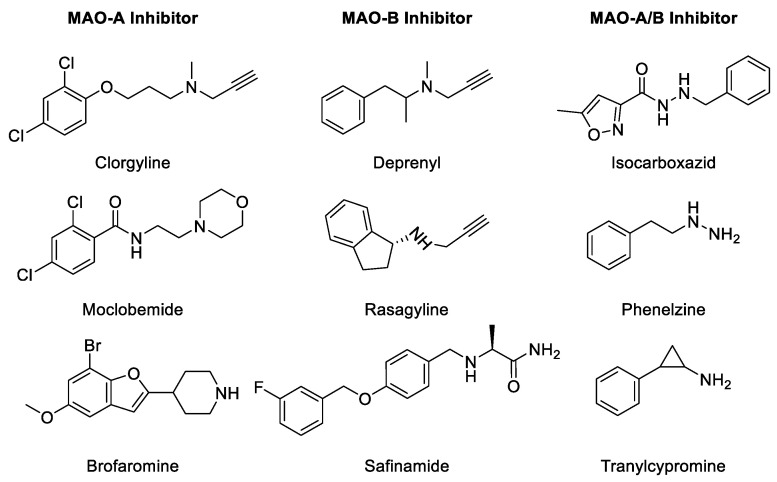
Chemical structure of selected MAO inhibitors.

**Figure 8 molecules-27-04297-f008:**
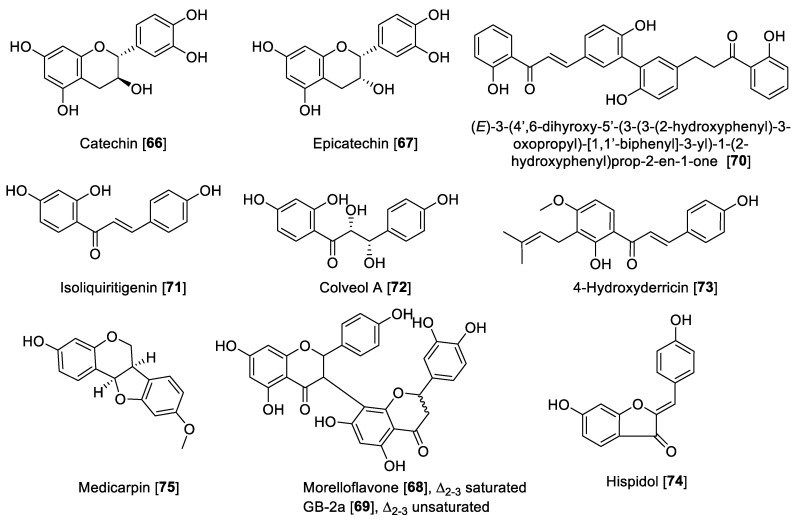
Chemical structure of selected miscellaneous flavonoid MAO inhibitors.

**Figure 10 molecules-27-04297-f010:**
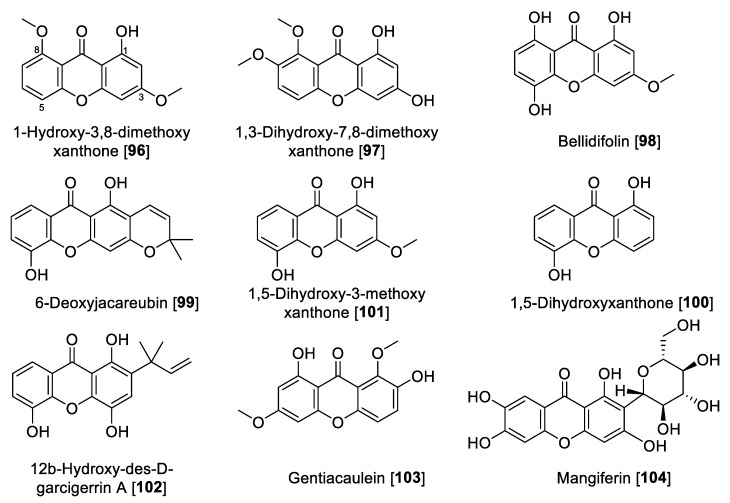
Chemical structure of selected xanthone MAO inhibitors.

**Figure 11 molecules-27-04297-f011:**
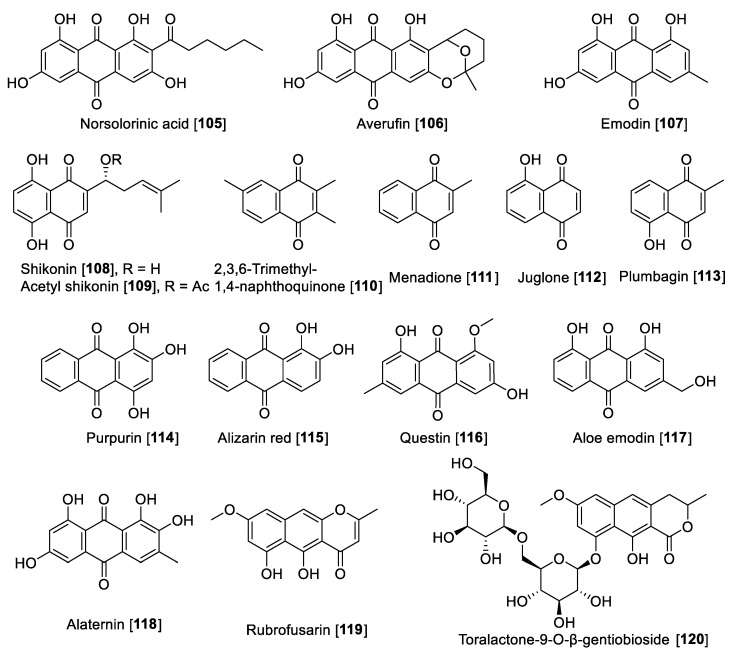
Chemical structure of selected naphthoquinone and anthraquinone MAO inhibitors.

**Figure 12 molecules-27-04297-f012:**
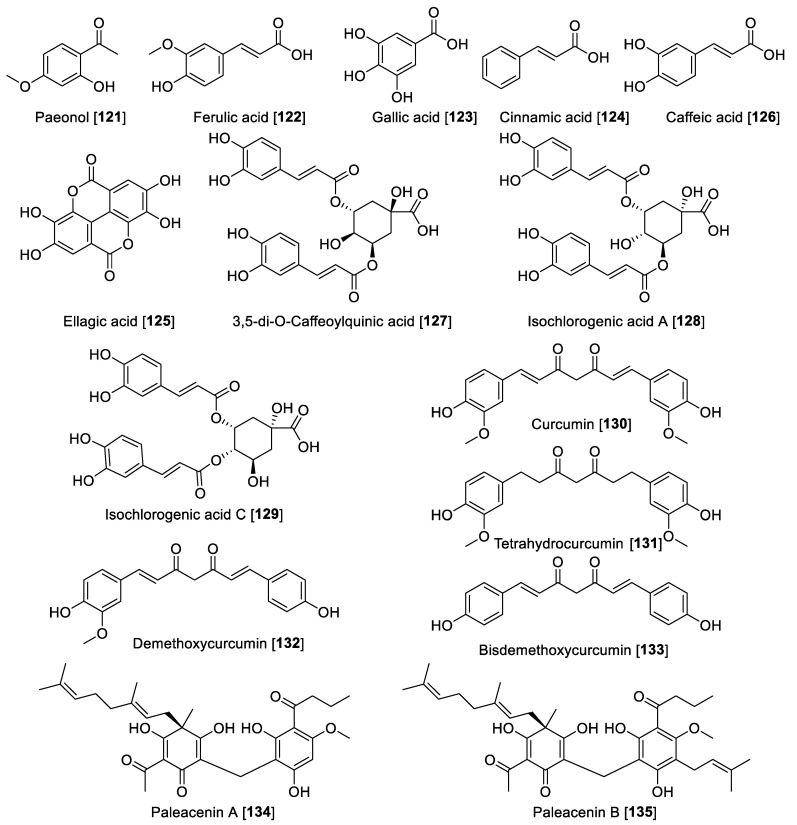
Chemical structure of miscellaneous polyphenol MAO inhibitors.

**Figure 13 molecules-27-04297-f013:**
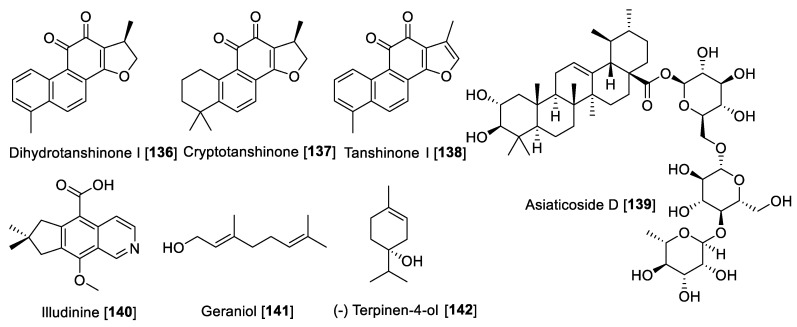
Chemical structure of selected terpenoid MAO inhibitors.

**Figure 14 molecules-27-04297-f014:**
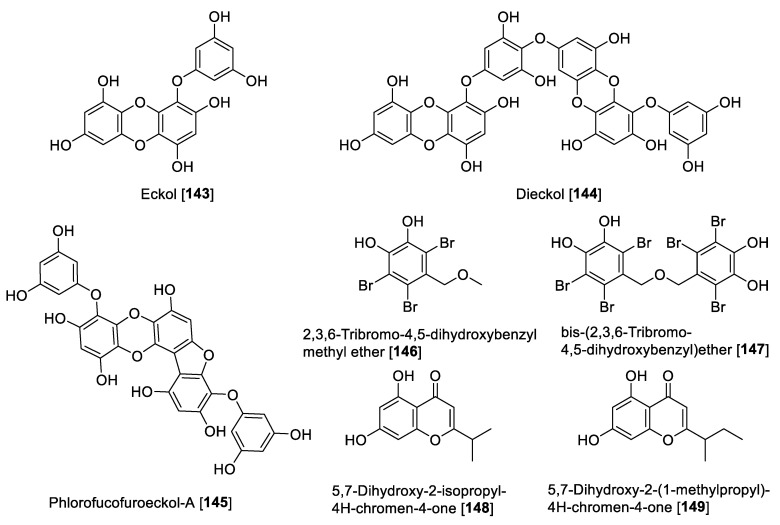
Chemical structure of selected marine isolated compound MAO inhibitors.

**Figure 15 molecules-27-04297-f015:**
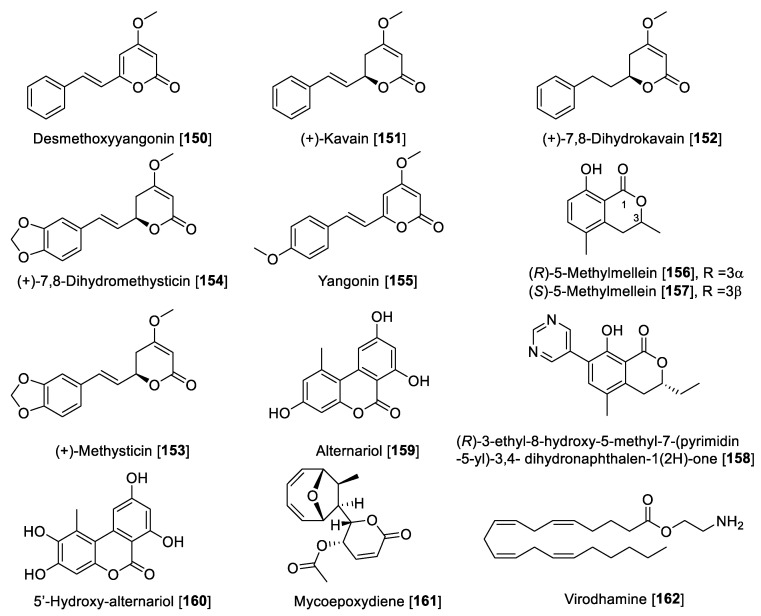
Chemical structure of selected miscellaneous compound MAO inhibitors.

## Data Availability

No experimental data are reported in the review manuscript.

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
