# Peer review of "Natural Products Inhibitors of Monoamine Oxidases—Potential New Drug Leads for Neuroprotection, Neurological Disorders, and Neuroblastoma"

_molecules, 2022, doi:10.3390/molecules27134297_

Round 1

Reviewer 1 Report

In this manuscript, Chaurasiya et al. reviewed natural products as MAO inhibitors. The manuscript is written well and comprehensively evaluates the raised issue.

There are several issues that I would like the authors to tackle:

  1. The chemical structure of all mentioned MAOIs, and drugs under development should be provided. Present also structure of natural MAOIs, and draw structure-activity relationships.
  2. A table summarizing IC50 concentrations of natural products would be an asset. Relate those concentrations to the approved drugs.
  3. Add a paragraph on the adverse effects of currently approved MAOIs.

Author Response

Reviewer 1 Comments and Suggestions for Authors

In this manuscript, Chaurasiya et al. reviewed natural products as MAO inhibitors. The manuscript is written well and comprehensively evaluates the raised issue.

There are several issues that I would like the authors to tackle:

1. The chemical structure of all mentioned MAOIs, and drugs under development should be provided. Present also structure of natural MAOIs, and draw structure-activity relationships.

Response. Chemical structures of therapeutic MAOIs have been added (Figure 3). The structures of natural products MAOIs presented in this review are already included in the manuscript.  SAR interferences with limited class of natural products where enough data are available have been presented in the individual sections on specific chemical class of natural products MAOIs    

2. A table summarizing IC50 concentrations of natural products would be an asset. Relate those concentrations to the approved drugs.

Response. The tables with summarized IC50 have been included for the revised manuscript .

3. Add a paragraph on the adverse effects of currently approved MAOIs.

Response. The section 3.1 (Line 177-219) presents the  limitations of currently approved MAOIs , which also includes adverse effects and potential interactions is already included in the manuscript  

Reviewer 2 Report

In my opinion, the submitted manuscript „Natural Products Inhibitors of Monoamine Oxidases- Potential New Drug Leads for Neuroprotection, Neurological Disorders and Neuroblastoma” meets aims and scope of „Molecules” Journal, section Natural Products Chemistry. The manuscript is full of errors and inaccuracies, therefore it may be accepted only after the major revision.

  1. According to the PRISMA guidelines for reviews (http://prisma-statement.org/PRISMAStatement/Checklist.aspx) authors should: „Specify the inclusion and exclusion criteria for the review and how studies were grouped for the syntheses. Specify all databases, registers, websites, organizations, reference lists and other sources searched or consulted to identify studies. Present the full search strategies for all databases, registers and websites, including any filters and limits used. Specify the methods used to decide whether a study met the inclusion criteria of the review, including how many reviewers screened each record and each report retrieved, whether they worked independently, and if applicable, details of automation tools used in the process…” The manuscript does not describe how the literature for this review was collected – it should be added (in the Introduction part, line 201), authors must underline the selective criteria and the restrictions for the literature search.
  2. The authors state that MAOI are the major class of drugs prescribed for treatment of depression and other neurological disorders (line 14). I do not agree with this opinion. There are at least 10 subgroups of tymoleptics (e.g. SSRI, SNRI, TCA, NARI…), and MAOI are just one of these subgroups. The similar sytuation is in line 67 (MAO inhibitors are a major class of medications – it's too much of a generalization).
  3. It is a fact that MAOI (selegiline) are used (and registered) in Parkinson’s disease treatment, but there are no MAOI registered to treat Alzheimer’s disease. To treat dementia there are only four small molecules: rivastigmine, galantamine, donepezil, memantine, and one biopharmaceutical – aducanumab. It is also known, that depression accompanies neurodegenerative disorders, that is why psychiatrists, when treating Alzheimer’s disease, very often, additionaly prescribe tymoleptics (sometimes also neuroleptics too, to treat the productive symptoms), but neither tymoleptics, nor neuroleptics are clasified as anti-dementia drugs. That is why, the authors are not allowed to claim that MAOI are use in the treatment of Alzheimer’s disease (line 81, 96, 190)! I am aware of the fact, that there are a lot of studies of various substances for treating neurodegenerative disorders (e.g. S. Manzoor, N.Hoda, A comprehensive review of monoamine oxidase inhibitors as Anti-Alzheimer’s disease agents: A review, E J Med Chem, Volume 206, 2020), but there is still no MAOI registered for such teatment, and it should be clearly explained in the manuscript.
  4. The statement: The most commonly used MAOIs are phenelzine, tricyclic antidepressants and benzodiazepines (line 138) is a nonsense. Neither TCAs, nor BZDs are MAOI. TCAs act mainly as SNRIs, they are also the antagonists of 5-HT receptors. BZDs dock to the GABA receptor and enhance the sedative effect of GABA on the central nervous system.
  5. Gingko biloba impoves cognitive thinking and memory, and may be additionaly helpful in early dementia not anxiety disorders as authors claim (line 183).
  6. In lines 244-246 the authors describe results for hMAO, please check it. In paper Mu, L. H.; Wang, B.; Ren, H. Y.; Liu, P.; Guo, D. H.; Wang, F. M.; Bai, L.; Guo, Y. S., Synthesis and inhibitory effect of piperine derivates on monoamine oxidase. Bioorg Med Chem Lett 2012, 22, (9), 3343‐8. authors claim that rat brain mitochondria were isolated from Sprague–Dawley rats according to the method of Clark and Nicklas.
  7. Evodia rutaecarpa is rather well-known in Chinese traditional medicine, than all over the world (line 263). Add the word Chinese.
  8. The authors claim that: The most important dietary sources of flavonoids are fruits, tea, and soybean (line 351). It is not true all over the world. There are countries, where people do not eat soybean (genistein), but drink a lot of red wine (resveratrol). Tea, as a beverage is popular, but sometimes, people drink only black tea, not as rich in flavonoids, as the green one is. In my opinion, a better statement would be: The most important dietary sources of flavonoids are green tea, fruits and vegetables.
  9. Gentiana lutea is a kind of a herbaceous plant, used as a medicinal plant. In medicine, the gentian root or rhizome is usually used, therefore the use of bark seems unlikely (line 460) – please check it (is it a bark from rhizome?).
  10. In Latin abbreviation „et al.” (from Latin: et alia) the dot after the word „al” is needed (e.g. line 569, 622).
  11. Numbers „0” and „1”  in the names of the stages of the cell cycle should be written in subscript (e. g. G0/G1 – line 808).
  12. Add the letter „f” in a word „figure” (line 224).
  13. Add space in line 443.
  14. With 160 chemical structures to be drawn, it is unlikely that you will not make a mistake. Other authors should have double checked the chemical structures and names in the publication intended to be a source of research for others. Please make the following corrections:
    1. the name of 4 should be: Piperic acid N-propyl amide; correct the chemical structure, because propyl has 3 carbons, not 2 as in the figure;
    2. the name of 12 should be: N-methyltetrandine chloride; + should refer to nitrogen, not hydrogen in this compound;
    3. the name of 13 should be: 1-methyl-2-lauryl-4(1H)-quinolone, n=10; lauric acid has 12 carbons, there are 12 carbons in the side chain of the drawn compound 13;
    4. the name of 26 should be: Tetrahydroharmine;
    5. correct the chemical structure of 59, there should be -OCH3 in 4’ position, not -OH;
    6. check the chemical structure of molecule 60, different sources give different structures (the substituent 5-methyl-2prop-1-en-2-ylhex-4-enyl is sometimes in position 6, sometimes in position 8 of benzo-γ-pyrone skeleton);
    7. the name of 70 should be: (E)-3-(4’,6-dihyroxy-5’-(3-(3-(2-hydroxyphenyl)-3-oxopropyl)-[1,1’-biphenyl]-3-yl)-1-(2-hydroxyphenyl)prop-2-en-1-one;
    8. check the chemical structure of molecule 68, there is –OH group missing in position 3 of B ring;
    9. there should be the lack of methyl group in 3’ position of 85 – correct the chemical structure;
    10. the name of 86 should be: 2-Methoxy-3-(1,1’-dimethylallyl)-6a,10a-dihydrobenzo(1,2-c)chroman-6-one;
    11. the name of 100 should be written without the space;
    12. the name of 115 should be „Alizarin red” to distinguish from another, because, there is also a compound named „Alizarin yellow”;
    13. check the chemical structure of molecule 118, -OH group should be in the 2. position (not 7) of anthracene skeleton;
    14. the name of 140 should be Illudinine, not Illudine;
    15. compounds 156 and 157, what does the „R=α, and R=β” mean? Is the carbon 4 in different configurations? How do we know what R refers to?
    16. A few chemical structures contain locants (e.g. 44, 96) and most do not. Be consistent.

Author Response

Reviewer - 2   Comments and Suggestions for Authors

In my opinion, the submitted manuscript „Natural Products Inhibitors of Monoamine Oxidases- Potential New Drug Leads for Neuroprotection, Neurological Disorders and Neuroblastoma” meets aims and scope of „Molecules” Journal, section Natural Products Chemistry. The manuscript is full of errors and inaccuracies, therefore it may be accepted only after the major revision.

  1. According to the PRISMA guidelines for reviews (http://prisma-statement.org/PRISMAStatement/Checklist.aspx) authors should: „Specify the inclusion and exclusion criteria for the review and how studies were grouped for the syntheses. Specify all databases, registers, websites, organizations, reference lists and other sources searched or consulted to identify studies.

Present the full search strategies for all databases, registers and websites, including any filters and limits used. Specify the methods used to decide whether a study met the inclusion criteria of the review, including how many reviewers screened each record and each report retrieved, whether they worked independently, and if applicable, details of automation tools used in the process…”

The manuscript does not describe how the literature for this review was collected – it should be added (in the Introduction part, line 201), authors must underline the selective criteria and the restrictions for the literature search.

Response.  The criteria and scope for survey and inclusion of the reports/data in the review have been included in the revised manuscript as below (Line 253-261).

“Electronic searches were specifically conducted on literature from December 2021 to April 2022 using major databases including PubMed, Google Scholar, SciFinder® and Web of Science. The keywords used in the searches were combination of words “natural prod-ucts’, “alkaloids”, “flavonoids”, “phenols”, “terpenes”, “monoamine oxidase inhibitors”, “MAO”, “MAOI”. We mostly selected published reports that reported the natural prod-ucts or natural products derivatives with inhibition of MAO-A or-B with IC50 <100 μM. These reports were further grouped into different chemical classes of natural product MAOIs. The natural products constituents tested against MAO-A and /or MAO-B were included in the data tables”

  1. The authors state that MAOI are the major class of drugs prescribed for treatment of depression and other neurological disorders (line 14). I do not agree with this opinion. There are at least 10 subgroups of tymoleptics (e.g. SSRI, SNRI, TCA, NARI…), and MAOI are just one of these subgroups. The similar sytuation is in line 67 (MAO inhibitors are a major class of medications – it's too much of a generalization).

Response. These sentences have been revised as “ Therapeutic MAOi are important class of drugs ……” (Abstract) and also as “The MAOIs are included in a group of drugs commonly referred as thymoleptic drugs that favorably modify mood in serious mood disorders and are primarily pre-scribed for the treatment of clinical depression or mania.” (Lines 123-125)

  1. It is a fact that MAOI (selegiline) are used (and registered) in Parkinson’s disease treatment, but there are no MAOI registered to treat Alzheimer’s disease. To treat dementia there are only four small molecules: rivastigmine, galantamine, donepezil, memantine, and one biopharmaceutical – aducanumab. It is also known, that depression accompanies neurodegenerative disorders, that is why psychiatrists, when treating Alzheimer’s disease, very often, additionally prescribe

Response. The mentions regarding therapeutic application of MAOIs for treatment of Alzheimer’s diseases have been deleted/revised. However, MAOI have been used off the label in the treatment of Alzheimer’s disease.

  1. tymoleptics (sometimes also neuroleptics too, to treat the productive symptoms), but neither tymoleptics, nor neuroleptics are clasified as anti-dementia drugs. That is why, the authors are not allowed to claim that MAOI are use in the treatment of Alzheimer’s disease (line 81, 96, 190)! I am aware of the fact, that there are a lot of studies of various substances for treating neurodegenerative disorders (e.g. S. Manzoor, N.Hoda, A comprehensive review of monoamine oxidase inhibitors as Anti-Alzheimer’s disease agents: A review, E J Med Chem, Volume 206, 2020), but there is still no MAOI registered for such teatment, and it should be clearly explained in the manuscript.

Response. To avoid misunderstandings, the mentions regarding therapeutic application of MAOIs for treatment of Alzheimer’s diseases have been deleted/revised

  1. The statement: The most commonly used MAOIs are phenelzine, tricyclic antidepressants and benzodiazepines (line 138) is a nonsense. Neither TCAs, nor BZDs are MAOI. TCAs act mainly as SNRIs, they are also the antagonists of 5-HT receptors. BZDs dock to the GABA receptor and enhance the sedative effect of GABA on the central nervous system.

Response. The statement has been revised  

  1. Gingko biloba improves cognitive thinking and memory, and may be additionally helpful in early dementia not anxiety disorders as authors claim (line 183).

Response. Early dem to include early dementia (Line 235)

  1. In lines 244-246 the authors describe results for hMAO, please check it. In paper Mu, L. H.; Wang, B.; Ren, H. Y.; Liu, P.; Guo, D. H.; Wang, F. M.; Bai, L.; Guo, Y. S., Synthesis and inhibitory effect of piperine derivates on monoamine oxidase. Bioorg Med Chem Lett 2012, 22, (9), 3343‐8. authors claim that rat brain mitochondria were isolated from Sprague–Dawley rats according to the method of Clark and Nicklas.

Response. This has been corrected (Line 300)

  1. Evodia rutaecarpa is rather well-known in Chinese traditional medicine, than all over the world (line 263). Add the word Chinese.

Response. The word Chinese has been added (Line 336).

  1. The authors claim that: The most important dietary sources of flavonoids are fruits, tea, and soybean (line 351). It is not true all over the world. There are countries, where people do not eat soybean (genistein), but drink a lot of red wine (resveratrol). Tea, as a beverage is popular, but sometimes, people drink only black tea, not as rich in flavonoids, as the green one is. In my opinion, a better statement would be: The most important dietary sources of flavonoids are green tea, fruits and vegetables.

Response. The phrase has been as  suggested by the reviewer (Line 421-422)

  1. Gentiana lutea is a kind of a herbaceous plant, used as a medicinal plant. In medicine, the gentian root or rhizome is usually used, therefore the use of bark seems unlikely (line 460) – please check it (is it a bark from rhizome?).

Response. The authors in PMID 15587710, did not clarify that, in the introduction they used dried bark and in the experimental used “wood”

  1. In Latin abbreviation „et al.” (from Latin: et alia) the dot after the word „al” is needed (e.g. line 569, 622).

Response. Corrected

  1. Numbers „0” and „1” in the names of the stages of the cell cycle should be written in subscript (e. g. G0/G1 – line 808).

Response. Corrected (line 909)

  1. Add the letter „f” in a word „figure” (line 224).

Response. Corrected

  1. Add space in line 443.

Response.  Corrected

  1. With 160 chemical structures to be drawn, it is unlikely that you will not make a mistake. Other authors should have double checked the chemical structures and names in the publication intended to be a source of research for others. Please make the following corrections:
    1. the name of 4 should be: Piperic acid N-propyl amide; correct the chemical structure, because propyl has 3 carbons, not 2 as in the figure;

Response. Corrected. All the chemical structures presented in this review have been thoroughly checked for accuracy. 

    1. the name of 12 should be: N-methyltetrandine chloride; + should refer to nitrogen, not hydrogen in this compound;

Response. The name of 12 has been mentioned as described in the reporte. We moved the plus up nitrogen to avoid misunderstandings.

    1. the name of 13 should be: 1-methyl-2-lauryl-4(1H)-quinolone, n=10; lauric acid has 12 carbons, there are 12 carbons in the side chain of the drawn compound 13;

Response. Thank you, n= is 10 because plus first methylene and plus the final methyl gave 12, same for the 13

    1. the name of 26 should be: Tetrahydroharmine;

Response. Corrected

    1. correct the chemical structure of 59, there should be -OCH3 in 4’ position, not -OH;

Response.  Corrected

    1. check the chemical structure of molecule 60, different sources give different structures (the substituent 5-methyl-2prop-1-en-2-ylhex-4-enyl is sometimes in position 6, sometimes in position 8 of benzo-γ-pyrone skeleton);

Response. Thank you we followed the chemical structure reported in PMID: 15789750

    1. the name of 70 should be: (E)-3-(4’,6-dihyroxy-5’-(3-(3-(2-hydroxyphenyl)-3-oxopropyl)-[1,1’-biphenyl]-3-yl)-1-(2-hydroxyphenyl)prop-2-en-1-one;

Response. Corrected

    1. check the chemical structure of molecule 68, there is –OH group missing in position 3 of B ring;

Response.  Corrected

    1. there should be the lack of methyl group in 3’ position of 85 – correct the chemical structure;

Response.  Corrected

    1. the name of 86 should be: 2-Methoxy-3-(1,1’-dimethylallyl)-6a,10a-dihydrobenzo(1,2-c)chroman-6-one;

Response. Corrected

    1. the name of 100 should be written without the space;

Response. Corrected

    1. the name of 115 should be „Alizarin red” to distinguish from another, because, there is also a compound named „Alizarin yellow”;

Response. Corrected

    1. check the chemical structure of molecule 118, -OH group should be in the 2. position (not 7) of anthracene skeleton;

Response. Corrected

    1. the name of 140 should be Illudinine, not Illudine;

Response. Corrected

    1. compounds 156 and 157, what does the „R=α, and R=β” mean? Is the carbon 4 in different configurations? How do we know what R refers to?

Response. It is clarified in the structures.

    1. A few chemical structures contain locants (e.g. 44, 96) and most do not. Be consistent.

Response. We included the number in the representative compounds as an illustration of the positions.

Reviewer 3 Report

The authors make a great effort to present a sprawling, dis-coordinated review of recent development of various types of naturally available MAO inhibitors for the treatment of neurological disorders and neuroblastoma. After reading, I believed that there is just too much work needed to make this a publishable and readable manuscript and cover what’s a rather complex, though worthwhile, topic. Therefore, I do NOT recommend this review to publish to Molecules. Here are just of list of a few (of many, many) issues:

  1. The authors listed various types of MAO inhibitors based on their skeletons, but not enough to satisfy the reader. In reviewer’s opinion, authors should list corresponding MAO A/B inhibitory activities under their chemical structures in all figures. Additionally, in order to stick to the main topic (New Drug Leads for Neuroprotection, Neurological Disorders and Neuroblastoma), the most potent MAOI(s) among each type which might be a lead or candidate, should provide more information (g. the MAO inhibition, PK, cellular and animal evaluation data, etc). I believed this information is more interesting and heuristic to readers, rather than simply listing different types of MAO inhibitors.
  2. The abstract part is poor-written; instead, concise abstract that can clearly express the main primary coverage of this review is strongly required.
  3. The different physiological and pathological roles, and also available crystal structures of two MAO subtypes, MAO-A and MAO-B, should be presented in more details. The diagrammatic presentations are suggested, which will make this review more readable and meaningful.
  4. In the ‘neuroblastoma’ portion, the chief biological data in vitro and in vivo should be added.
  5. I believed this information is more interesting and heuristic to readers, rather than simply listing different types of inhibitors.
  6.  Other queries: the tile orders should use 1.2.3……rather than 1.0, 2.0, 3.0….

Author Response

Reviewer 3 Comments and Suggestions for Authors

The authors make a great effort to present a sprawling, dis-coordinated review of recent development of various types of naturally available MAO inhibitors for the treatment of neurological disorders and neuroblastoma. After reading, I believed that there is just too much work needed to make this a publishable and readable manuscript and cover what’s a rather complex, though worthwhile, topic. Therefore, I do NOT recommend this review to publish to Molecules. Here are just of list of a few (of many, many) issues:

  1. The authors listed various types of MAO inhibitors based on their skeletons, but not enough to satisfy the reader. In reviewer’s opinion, authors should list corresponding MAO A/B inhibitory activities under their chemical structures in all figures. Additionally, in order to stick to the main topic (New Drug Leads for Neuroprotection, Neurological Disorders and Neuroblastoma), the most potent MAOI(s) among each type which might be a lead or candidate, should provide more information (g. the MAO inhibition, PK, cellular and animal evaluation data, etc). I believed this information is more interesting and heuristic to readers, rather than simply listing different types of MAO inhibitors.

Response. Currently, harmine 22 is subjected to a clinical trial in combination with DMT 28, to study the network dynamics following the modulation of the serotonin system the point to be added in conclusion.

  1. The abstract part is poor-written; instead, concise abstract that can clearly express the main primary coverage of this review is strongly required.

Response- The abstract has been revised as suggested 

  1. The different physiological and pathological roles, and also available crystal structures of two MAO subtypes, MAO-A and MAO-B, should be presented in more details. The diagrammatic presentations are suggested, which will make this review more readable and meaningful.

Response. Section 2 was added about structural differential of MAOA and B.

  1. In the ‘neuroblastoma’ portion, the chief biological data in vitro and in vivo should be added. I believed this information is more interesting and heuristic to readers, rather than simply listing different types of inhibitors.

Response- Current data on therapeutic potential of natural products MAOIs for neuroblastoma are limited. Therapeutic potential of harmala alkaloids such as harmine 22 and norharman 27 (the MAO-A inhbitors ) has been described in enough details (Line 974-985). The key results with berberine have also been included (Line 986-991). Even though current data on this are limited, the section has been included in this review to draw attention of the researchers towards potential therapeutic application of natural products MAOIs for neuroblastoma and promote future research in this area.        

  1.  Other queries: the tile orders should use 1.2.3……rather than 1.0, 2.0, 3.0…

Response. Changed 1.2. and 3.

Reviewer 4 Report

The article titled Natural Products Inhibitors of Monoamine Oxidases‐ Potential New 2 drug Leads for Neuroprotection, Neurological Disorders and Neuro‐ 3 Blastoma after consideration of the following  comments

  • Abstract, authors should not use we , us I etc. 
  • Abstract , should contain brief conclusion.
  • Introduction, line 224 change igures to figures.
  • It is better for authors to add natural FDA approver MAOI
  • Also, it is better to make SAR for each phytochemical class and detect the common SAR for all classes
  • I suggest mentioning the approved drugs SAR and corelated with active natural metabolites.
  • I suggest to support this study with some docking studies.
  • Conclusions need to improve and be concise 

The study is very important and concerned on new subject, I think it is very useful for design of new drug development.

Author Response

Reviewer - 4 Comments and Suggestions for Authors

The article titled Natural Products Inhibitors of Monoamine Oxidases‐ Potential New 2 drug Leads for Neuroprotection, Neurological Disorders and Neuro‐Blastoma after consideration of the following comments.

  • Abstract, authors should not use we, us, I etc. 

Response. The abstract has been revised  

  • Abstract, should contain brief conclusion.

Response- The last few sentences included in the abstract present important conclusions

  • Different classes of MAOI have been identified from the natural products sources with non-selective as well as selective inhibition of MAO- A and -B.
  • Selective natural product MAOI may be safer alternatives to the conventional MAOI drugs.
  • Characterization of MAO inhibitory constituents of natural products traditionally used as psychoactive preparations or for treatment of neurological disorder will help in understanding the mechanism of action and optimization of these preparations for desired bioactive properties.
  • Potential therapeutic application of natural products MAOIs for treatment of neuroblastoma has also been discussed.

The last sentence on potential herb-drug interactions of natural products MAOIs has been deleted, since this topic has not been covered in this review and may require separate attention.   

  • Introduction, line 224 change igures to figures.

Response. Corrected

  • It is better for authors to add natural FDA approver MAOI

Response. A section on therapeutic applications of MAOIs (Section 2.2) is already included. At this time no natural product MAOI has been approved by the FDA.   

  • Also, it is better to make SAR for each phytochemical class and detect the common SAR for all classes

Response. SAR for selected compounds (flavonoids/alkaloids) group of series added.

  • I suggest mentioning the approved drugs SAR and corelated with active natural metabolites.

Response- A section on “Therapeutic applications of MAO inhibitors” (Section 2.2) is already included. SAR inferences has been presented with selected classes of natural products MAOIs, where enough data are available 

  • I suggest to support this study with some docking studies.

Response- Docking studies are beyond the scope of this review and may be included in future reports. The studies are in progress with selected classes on natural products MAOIs.  

  • Conclusions need to improve and be concise 

Response- Conclusions have been revised to present key results of this review.    

Round 2

Reviewer 1 Report

I do not have any more comments.

Author Response

Reviwer-1.

Response – Thanks for your valuable response and Don’t have further comments.

 (x) Moderate English changes required
 Response-Thoroughly checked and revised english grammar.

Reviewer 2 Report

In my opinion, the submitted manuscript „Natural Products Inhibitors of Monoamine Oxidases- Potential New Drug Leads for Neuroprotection, Neurological Disorders and Neuroblastoma” may be accepted in present form.  Thank you for responding to my comments and introducing corretions to the manuscript.

Author Response

Thanks for your valuable response and Don’t have further comments on revised manuscript.

(x) I don't feel qualified to judge about the English language and style

Response-Thoroughly checked and revised english grammar.

Reviewer 3 Report

Minor modifications need to be done, such as the correct expression of IC50 (line 472, 482, etc), Ki (line 277)  

Author Response

Comment- Minor modifications need to be done, such as the correct expression of IC50 (line 472, 482, etc), Ki (line 277)  

Response. The IC50 expression has been changed as suggested by the reviewer (Line 472, 482 et. and Ki in line 277.

(x) Moderate English changes required

Response – Thanks for your valuable comments and we have thoroughly checked and revised english grammar.